# Chemical tools to define and manipulate interferon-inducible Ubl protease USP18

Griffin J. Davis[1,5], Anthony O. Omole [1,5], Yejin Jung[1,5], Wioletta Rut[2], Ronald Holewinski[3], Kiall F. Suazo [3], Hong-Rae Kim[1,4], Mo Yang[1], Thorkell Andresson[3], Marcin Drag [2] & Euna Yoo [1] ✉

Ubiquitin-specific protease 18 (USP18) is a multifunctional cysteine protease primarily responsible for deconjugating the interferon-inducible ubiquitin-like modifier ISG15 from protein substrates. Here, we report the design and synthesis of activity-based probes (ABPs) that incorporate unnatural amino acids into the C-terminal tail of ISG15, enabling the selective detection of USP18 activity over other ISG15 cross-reactive deubiquitinases (DUBs) such as USP5 and USP14. Combined with a ubiquitin-based DUB ABP, the USP18 ABP is employed in a chemoproteomics screening platform to identify and assess inhibitors of DUBs including USP18. We further demonstrate that USP18 ABPs can be utilized to profile differential activities of USP18 in lung cancer cell lines, providing a strategy that will help define the activity-related landscape of USP18 in different disease states and unravel important (de)ISGylation-dependent biological processes.

Deubiquitinases (DUBs), a family of proteases that cleave ubiquitin-like molecules (Ubls) from their substrate proteins, play critical roles in regulating protein turnover and non-degradative functions such as DNA repair, protein complex formation, cellular trafficking, localization, and inflammation[1]. DUBs are broadly implicated in many pathophysiological conditions including cancer given their fundamental roles in various cellular processes[2].

Among ~100 reported mammalian DUBs, ubiquitin-specific protease 18 (USP18) is primarily responsible for the cleavage of the interferon (IFN)-inducible Ubl, interferon-stimulated gene 15 (ISG15) with exquisite specificity[3]. In response to various cellular stresses, particularly viral infections and other immune stimuli, ISG15 conjugation (ISGylation) is mediated by the consecutive action of a three-step catalytic cascade in a similar manner to ubiquitylation[4,5], and conversely counteracted by USP18 isopeptidase activity (Fig. 1A). ISGylation of over 300 proteins has been reported in IFN-stimulated cells, including proteins involved in innate immune response as well as oncogenic and tumor-suppressive proteins, without a clear consensus sequence[6–10]. The ISG15 system has been largely implicated in antiviral immunity as an important cellular defense mechanism to restrict infections since elevated ISGylation of viral or host proteins has been reported to accompany increased viral resistance[11–13]. There is also mounting evidence that ISGylation plays important roles in maintaining genome stability, cytoskeleton dynamics, autophagy, protein translation, and hypoxia/ischemic responses[14]. While much attention has been given to the function of ubiquitin in protein degradation, it remains unclear how protein ISGylation is associated with protein stability, degradation, or cross-talks with ubiquitylation. Independent of its deISGylating activity, USP18 also binds to IFN-α/β receptor 2 complex, where it competes with JAK1, thereby negatively regulating type I IFN signaling[15,16]. mRNA and protein levels of USP18 are found to be elevated in diverse cancers and USP18 loss has been shown to restrict tumor growth[17,18]. Despite these findings, the mechanistic contribution of this multifaceted protein to cancer development, progression, or survival remains largely understudied. USP18 can modulate the stability, localization, and functional activity of proteins

[1]Chemical Biology Laboratory, Center for Cancer Research, National Cancer Institute, National Institutes of Health, Frederick, MD, USA. [2]Department of Chemical Biology and Bioimaging, Wroclaw University of Science and Technology, Wroclaw, Poland. [3]Protein Characterization Laboratory, Frederick National Laboratory for Cancer Research, Leidos Biomedical Research, Frederick, MD, USA. [4]Present address: Department of Biomedical Sciences, College of Medicine, Korea University, Seoul, South Korea. [5]These authors contributed equally: Griffin J. Davis, Anthony O. Omole, Yejin Jung. ✉e-mail: euna.yoo@nih.gov

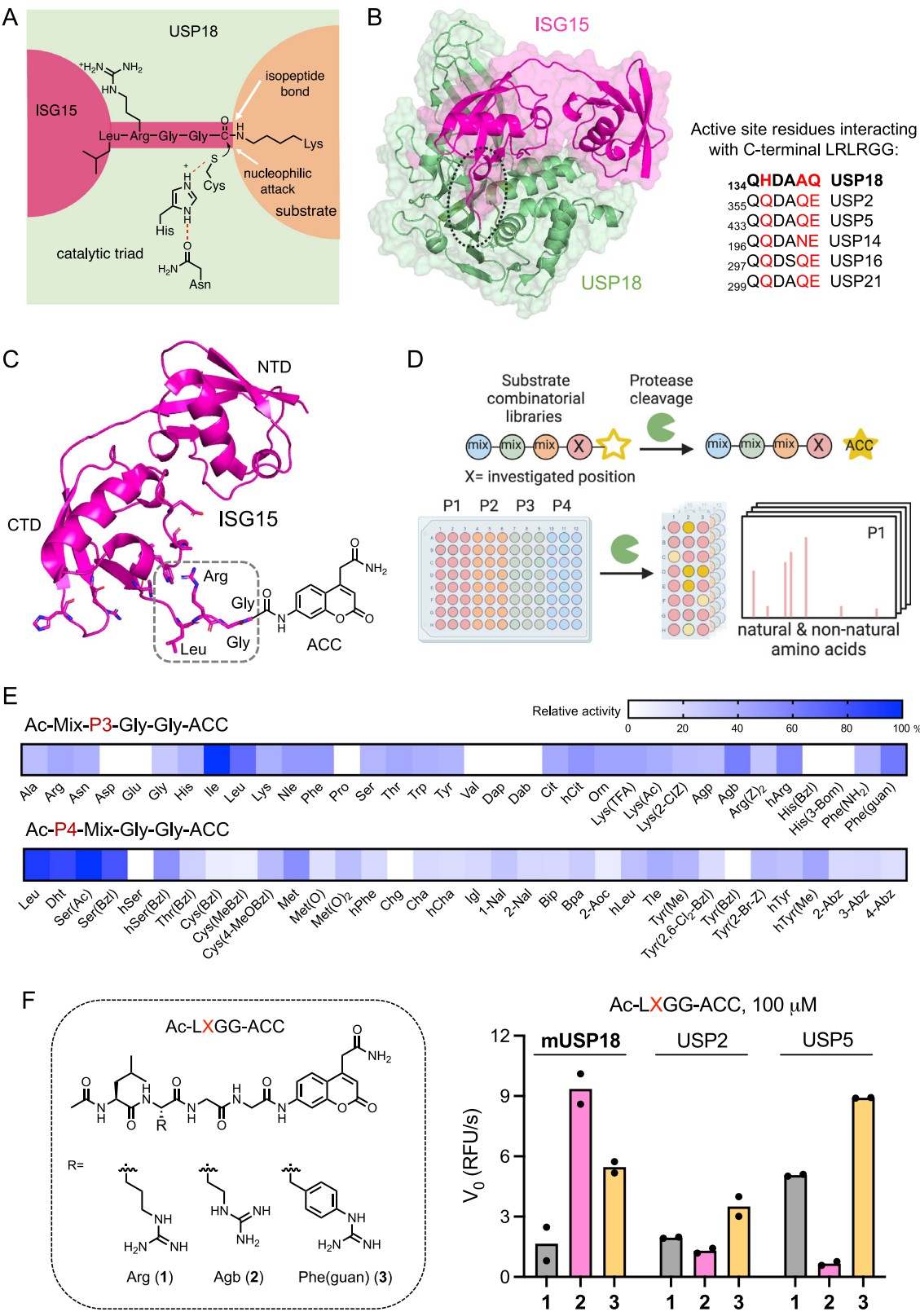

**E**

Ac-Mix-P3-Gly-Gly-ACC

Relative activity

0  20  40  60  80  100 %

Ac-P4-Mix-Gly-Gly-ACC

**F**

Ac-LXGG-ACC

R=

Arg (**1**)   Agb (**2**)   Phe(guan) (**3**)

Ac-LXGG-ACC, 100 μM

important to tumor growth and antitumor immunity by regulating protein deISGylation and protein-protein interactions[19–23]. As a major regulator of the IFN signaling network and T cell differentiation, USP18 can also affect inflammatory and immune responses within the tumor immune microenvironment[24–26]. The role that DUBs play in cancer is often complex and highly context-dependent. Similarly, depending on the substrate landscape, subcellular localization, cell type, and

physiologic states, each cancer type may respond differently to USP18 loss or inhibition. In addition, there are multiple mechanisms including protein binding and post-translational modifications (PTMs) that modulate DUB activity, the dysregulation of which has been observed in many cancers[27–29]. Recently, integrated multi-omics data revealed that the sensitivity of individual tumor lineages to different DUB knockouts is often not correlated with protein abundance[30],

**Fig. 1 | Fluorogenic substrates for USP18. A** Catalytic triad of USP18. The iso-peptide bond between the C-terminal Gly residue of ISG15 and the Lys residue of the substrate protein is hydrolyzed by the USP18 protease activity. **B** Structure of USP18 complexed with ISG15 (PDB: 5CHV). Active site residues of USP18 that interact with conserved C-terminal LRLRGG of ISG15 are highlighted in comparison to other ISG15 reactive DUBs. **C** Structure of ISG15-based ACC-labeled substrate. The C-terminal LRGG motif is highlighted. **D** Schematic overview of HyCoSuL screening created in BioRender. Yoo, E. (2025) https://BioRender.com/z61a243. **E** Substrate profiles presented as heatmaps indicating the percentage of relative cleavage measured by a fluorescent signal produced upon USP18 binding with two sub-libraries, Ac-Mix-P3-Gly-Gly-ACC and Ac-P4-Mix-Gly-Gly-ACC (where Mix represents an equimolar mixture of natural amino acids). Source data are provided as a Source Data file. **F** Structures (left panel) and rate of hydrolysis (right panel) of selected tetrapeptide fluorogenic substrates. mUSP18 (10 μM), USP2 (2 μM), and USP5 (1 μM) were incubated with 100 μM of Ac-LXGG-ACC substrates for 1 h at RT and initial release of fluorescent ACC ($V_0$, RFU/s) by enzyme was measured at Ex: 360 nm / Em: 460 nm. Agb 2-amino-4-guanidino-butyric acid, hArg homoarginine, Phe(guan) guanidino-phenylalanine. Data represent mean values ($n = 2$ independent replicates). Source data are provided as a Source Data file.

underscoring the importance of understanding the precise roles that each DUB plays. There is a significant need for the development of chemical tools that allow the study activity of USP18 in health and disease conditions in addition to measuring expression levels, which will also provide insight into USP18-targeting therapeutic strategies.

A chemoproteomics method termed activity-based protein profiling (ABPP) that uses activity-based probes (ABPs) is a powerful platform to map proteome-wide reactive proteins and evaluate functional states of enzymes in complex biological systems[31–35]. Using ABPs that mimic native ubiquitin but are modified with an electrophilic warhead at the C-terminus to label the active site cysteines irreversibly, ABPP has led to the identification of new DUB families and characterization of DUB activity across conditions[36]. In a recent study, an ISG15-based ABP was used to detect deISGylating enzymes in human cell lysates, identifying USP16 as an ISG15 cross-reactive DUB[37] in addition to previously-characterized DUBs such as USP5 and USP14[38–40]. Competitive ABPP has also been implemented to screen DUB inhibitors[41,42].

Here, we report the design and synthesis of ABPs that more selectively detect deISGylase activity of USP18. Through screening of a hybrid combinatorial substrate library (HyCoSuL)[43], we identified mutations in the LRGG tail of ISG15 that enhance selective binding to USP18 over other ISG15 cross-reactive DUBs. We incorporated these mutations into a C-terminal domain of murine ISG15 using a total chemical synthesis approach and evaluated their efficiency and selectivity in cell lysate conditions. The USP18 ABPs with improved selectivity were then employed to identify WP1130 as an inhibitor of USP18 in a workflow that can be adapted as a chemoproteomics screening platform for discovering and characterizing DUB inhibitors. We further utilized them for profiling the USP18 activity in cell lines derived from lung carcinoma. Our study indicates that the differential activity of USP18 can be measured using this approach to investigate its activity-related biological functions.

## Results

### Design and synthesis of mISG15$_{CTD}$-based probes for selective detection of USP18

In general, a DUB ABP consists of three components: a ubiquitin-like protein (recognition element), an electrophile such as a vinyl sulfone (VS) or propargyl (PA) group (reactive warhead), and a fluorophore/affinity handle (reporter tag)[36]. Upon Ubl binding to a DUB, the electrophile is covalently attached to the active site cysteine in an enzyme-catalyzed reaction, and probe labeling thus reports DUB activity. To develop an ABP for USP18, we first explored the ISG15 sequence as a recognition element. ISG15 consists of two Ubl domains connected by a short hinge region and contains the canonical LRGG motif at its C-terminus, which is required for conjugation to its targets[44]. When we tested monoUb, ISG15 (ISG15$_{FL}$), and the C-terminal Ubl domain of ISG15 (ISG15$_{CTD}$) in vitro, we observed that USP18 prefers and efficiently binds to ISG15 over Ub (as previously reported) and ISG15$_{CTD}$ is sufficient for binding to USP18 (Supplementary Fig. 1A)[45]. We also found that mouse ISG15 (64% sequence identity to human ISG15) binds well to human USP18. When native ISG15 is used as a recognition element, however, these ISG15-based probes have been reported to label

not only USP18 but also other ISG15 cross-reactive DUBs such as USP2, USP5, USP14, and USP16[38,39]. We found that USP5 was indeed labeled by both hISG15$_{FL}$ and mISG15$_{CTD}$-based probes (Supplementary Fig. 1B).

To develop an ABP that specifically reports on deISGylating activity of USP18, we hypothesized that creating mutations in certain regions of ISG15, while keeping key residues that confer specificity over Ub[45], may optimize intermolecular contacts and direct it toward USP18 with enhanced binding affinity and selectivity. To assess whether mutations in the conserved C-terminal LRGG motif can be tolerated or even enhance interaction with USP18 based on subtle differences in the active site residues of USP18 compared to other ISG15 cross-reactive DUBs (Fig. 1B), we performed positional scanning of a hybrid combinatorial substrate library (HyCoSuL)[43,46], which allows rapid substrate profiling for amino acid preferences including natural and unnatural amino acids at the sites adjacent to the scissile bond (Fig. 1C, D). P3 and P4 sub-libraries of fluorogenic tetrapeptide substrates that bear Gly residues at the P1 and P2 and a 7-amino-4-carbamoylmethylcoumarin (ACC) fluorophore at the P1' position were screened with USP18. We found that USP18 recognizes and cleaves tetrapeptide substrates that contain unnatural amino acids, for example, Agb, hArg, Phe(guan) at the P3, and Dht, Ser(Ac), Ser(Bzl) at the P4 position (Fig. 1E). We then synthesized seven individual fluorogenic substrates (i.e., Ac-LXGG-ACC and Ac-XRGG-ACC) and measured the rate of hydrolysis (Supplementary Fig. 2). Ac-Leu-Agb-Gly-Gly-ACC substrate (**2**) was cleaved faster than Ac-LRGG-ACC (**1**) by USP18 but showed decreased rates of cleavage by USP2 and USP5, while Ac-Leu-Phe(guan)-Gly-Gly-ACC substrate (**3**) was efficiently cleaved by all three enzymes (Fig. 1F). This suggests that USP18 can recognize amino acids other than the canonical Leu-Arg in the P4-P3 positions, while replacing the P3 Arg with Agb might enhance selectivity towards USP18.

Since tetrapeptide sequences are not sufficient to fully engage USP18 thus requiring high concentrations, we decided to synthesize the C-terminal domain of mISG15 with the identified mutations as we found mISG15$_{CTD}$ has sufficient binding to both human and mouse USP18 and is more synthetically manageable than hISG15$_{CTD}$. Using standard solid-phase peptide synthesis, we incorporated unnatural amino acids, warheads, and reporter tags in a straightforward manner with chemical flexibility (Fig. 2A). Briefly, we used Fmoc-Gly TentaGel Trt resin and sequentially deprotected Fmoc and coupled following amino acids in the presence of PyBOP and DIPEA in NMP[47]. To improve stability and avoid dimerization, Cys144 was replaced by Ser. Mutant probes were generated by substituting Arg153 with Agb, hArg, or Phe(guan). Once mISG15$_{CTD}$ (Leu80-Gly154) was assembled, the N-terminal end was either acetylated or conjugated with an analytical handle (a Cy5 fluorophore for visualization or biotin for affinity purification). After cleavage from the solid support, the C-terminal end was further functionalized with a propargylamine warhead, replacing Gly155, for covalent conjugation[48]. The identity and purity of each probe were confirmed by LC-MS analysis (Fig. 2B and Supplementary Information). Our SPR analysis of Ac-mISG15$_{CTD}$-OH indicated that both Agb and Phe(guan) mutants retain binding affinities toward USP18 that are comparable to the wild type (Supplementary Fig. 3).

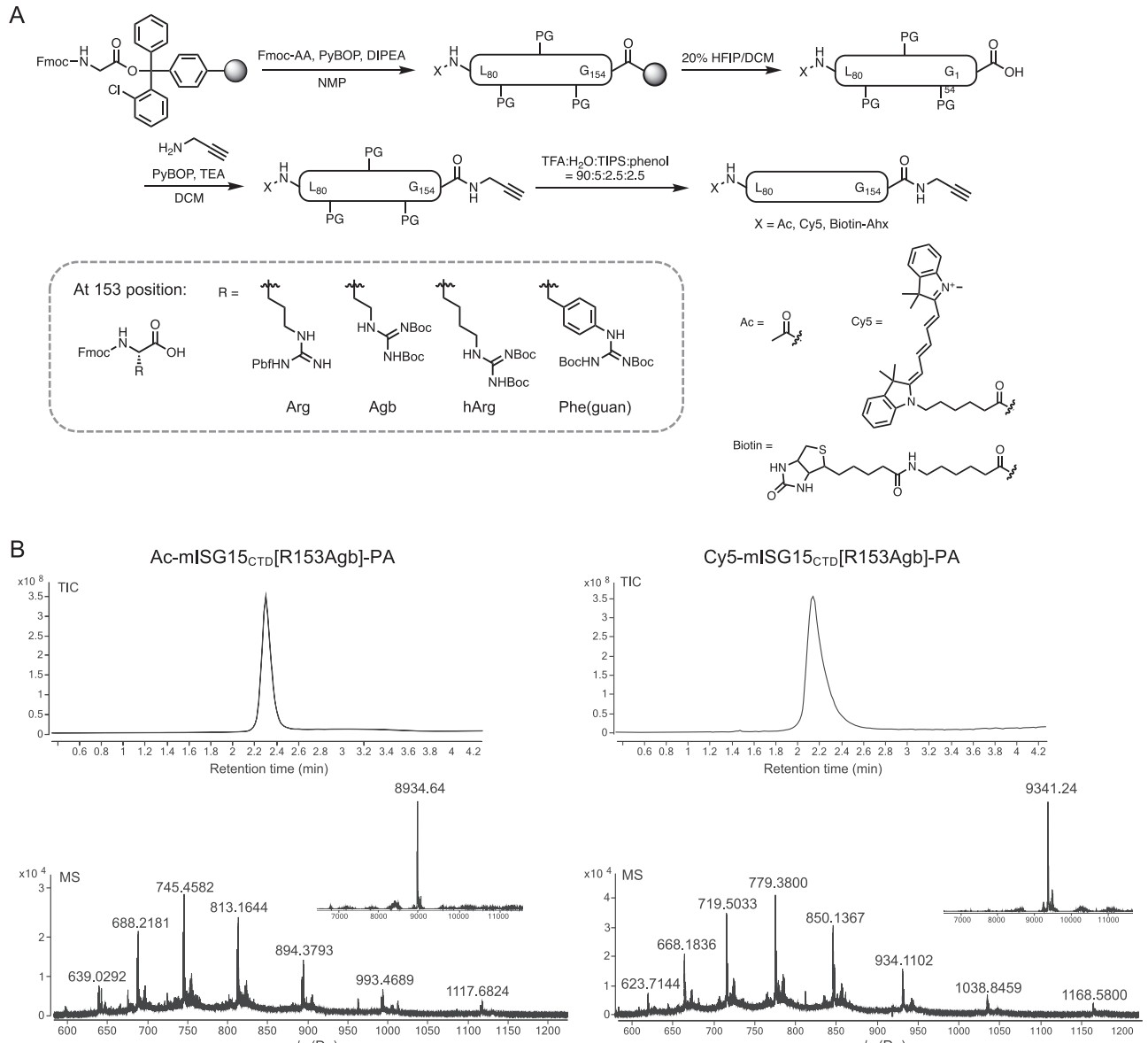

**Fig. 2 | Preparation of mISG15<sub>CTD</sub>-based probes. A** Synthesis of C-terminal domain of mouse ISG15 and ISG15 variants. Leu80-Gly154 sequence was assembled using SPPS and the N-terminal end was modified by either an acetyl group, fluorescent cyanine dye, or biotin affinity handle. After the cleavage from the solid support, the C-terminal end was functionalized with propargylamine (PA) replacing Gly155. Arg153 was varied with unnatural amino acids such as Agb (2-amino-4-guanidino-butyric acid), hArg (homoarginine), Phe(guan)(guanidino-phenylalanine). **B** LC-MS analysis of synthetic Ac-mISG15<sub>CTD</sub>-PA and Cy5-mISG15<sub>CTD</sub>-PA probes.

## Reactivity and selectivity assessment of mISG15<sub>CTD</sub>-based probes

We then performed gel-based labeling assays to assess the reactivity and selectivity of mutant ISG15 probes. When incubated with either mouse or human recombinant USP18, all tested Ac-mISG15<sub>CTD</sub>-PA probes showed efficient labeling of the active enzyme, judging from the appearance of a distinct, higher molecular weight protein band owing to the covalent bond formation (Fig. 3A). Importantly, the Agb and Phe(guan) mutant probes showed diminished to marginal interaction with USP5, while the wild type (WT) and hArg mutant displayed detectable conjugation (Fig. 3B), revealing that subtle change in the tail of ISG15, particularly at the 153 position (varying the length of the side chain by no more than two methylene carbons), has significant consequences for its molecular interaction.

To visualize DUB–probe conjugates in complex proteomes, we first used Cy5-Ubl-PA probes in lysates from HEK293T cells

(Supplementary Fig. 4). Additionally, we included an H90F mutant of mISG15<sub>CTD</sub> (a mutation outside of the tail region) that was computationally predicted to increase binding to USP18 through hydrophobic interactions according to structure-based computational mutational analysis utilizing FoldX software[49] and RosettaDesign server[50]. Through in silico alanine and positional scanning using the crystal structure of the mUSP18-ISG15 complex (PDB ID: 5CHV)[45], we identified key interface residues that optimize the interaction between ISG15 and USP18. In our labeling experiments, the Ub probe demonstrated clear labeling of many proteins, while the WT mISG15<sub>CTD</sub> probe showed strong labeling of fewer DUBs, such as USP5, USP16, and USP14 based on the molecular weight, as expected. With no detectable basal expression of USP18, Arg153Agb and Arg153Phe(guan) mutant probes showed decreased background labeling relative to the WT, whereas the H90F mutant probe showed increased background labeling as noted by a more intense band with a molecular weight near 76 kDa

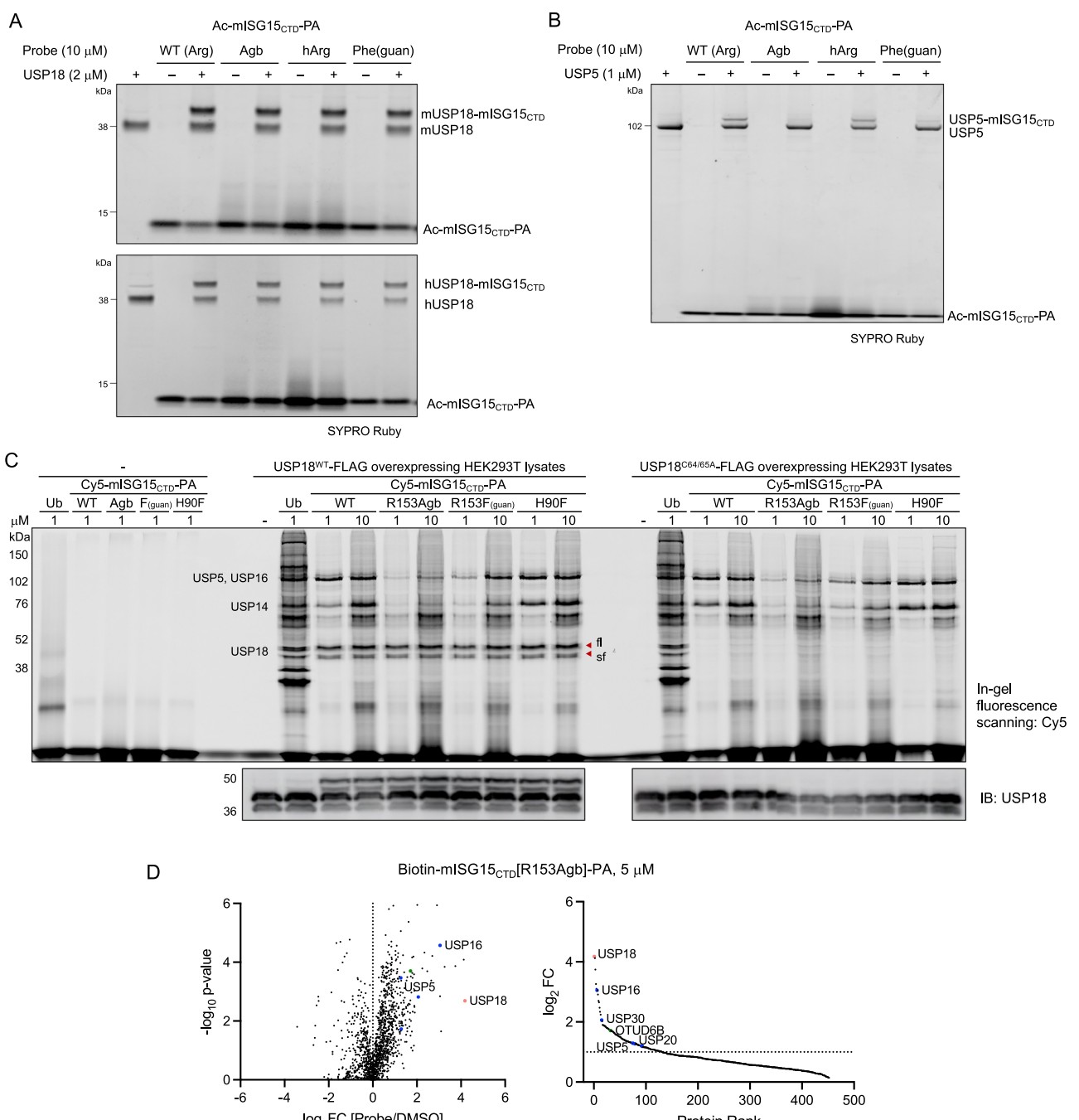

**Fig. 3 | Reactivity and selectivity of mISG15_CTD-based probes. A**, **B** Recombinant mouse and human USP18 (2 μM) or USP5 (1 μM) were incubated with Ac-mISG15_CTD-PA probes (10 μM) for 3 h at RT. Protein samples were analyzed by SDS-PAGE and SYPRO Ruby staining. The result is a representative of three experiments ($n = 3$ independent replicates). Source data are provided as a Source Data file. **C** WT or C64/65A mutant of USP18-FLAG overexpressing HEK293T cell lysates were incubated with Cy5-mISG15_CTD-PA probes at indicated concentrations for 3 h at RT. Protein samples were analyzed by SDS-PAGE and in-gel fluorescence scanning for Cy5 signal. Red marks indicate the appearance of new protein bands only in USP18^WT overexpressing cell lysates after labeling by probes corresponding to the expected molecular weight of USP18-probe conjugate. Expression of USP18 was

confirmed by western blotting. The result is a representative of three experiments ($n = 3$ independent replicates). Source data are provided as a Source Data file. **D** Volcano (left) and protein rank (right) plots of quantitative proteomic analysis of streptavidin beads pulldowns after labeling of USP18-FLAG overexpressing HEK293T cell lysates by Biotin-mISG15_CTD[R153Agb]-PA probe (5 μM, 3 h, RT) showing significantly enriched proteins (log$_2$ ratio >1, $p$-value ≤ 0.05). USP18 is marked red and other DUBs are colored based on subfamilies (blue: USP, red: UCH, green: OTU, and cyan: JAMM). Data represent mean values ($n = 3$ independent replicates). $p$-values were calculated by two-tailed $t$-test. Source data are provided with this paper (Supplementary Data 2).

(Supplementary Fig. 4). Performing the same labeling experiment in USP18^WT overexpressing HEK293T cell lysates, two additional protein bands appeared corresponding to covalent adducts of two USP18 isoforms[51] (red arrows in Fig. 3C), which was elaborated by western blotting analysis where both unconjugated and conjugated full length

(fl) and truncated (sf) USP18 were detected. Satisfyingly, when the labeling assay was performed with lysates of HEK293T cells overexpressing catalytically inactive USP18^C64A/C65A (a double mutation of catalytic (C64) and adjacent (C65) cysteines)[21], there was no USP18–probe conjugate detected, confirming that apparent labeling

of USP18 by ABPs is through covalent conjugation in a catalytic cysteine-dependent manner. Overall, the Arg153Agb mutant appeared to be slightly more selective for USP18 than the Arg153Phe(guan) mutant, as concluded from less intense labeling of other ISG15 cross-reactive DUBs at both 1 μM and 10 μM. From USP18[WT] overexpressing HEK293T cell lysates preincubated with either the WT or Arg153Agb mutant of Ac-mISG15$_{CTD}$-PA prior to labeling with Cy5-mISG15$_{CTD}$[WT]-PA, we observed that the R153Agb mutant selectively blocked USP18 labeling while the WT probe competed for all the other DUBs in addition to USP18 (Supplementary Fig. 5), again confirming enhanced selectivity of Arg153Agb for USP18 compared to the WT.

For parallel affinity pulldown and LC-MS/MS analysis, we utilized biotin ABPs. Biotin probes were first evaluated with recombinant enzymes to ensure that N-terminal modification with biotin retains the same behavior of binding and specificity. The biotin-mISG15$_{CTD}$[R153Agb]-PA probe showed sufficient binding to USP18 (Supplementary Fig. 6A) with minimal binding to USP5 (Supplementary Fig. 6B). Immunoblotting analysis of pulldown experiments using USP18 overexpressing HEK293T cell lysates confirmed enrichment of active USP18 by the R153Agb mutant probe with no or minimal pull-down of USP14 or USP5, respectively (Supplementary Fig. 6C). The probe-labeled, streptavidin-enriched proteins were then subjected to on-bead digestion, isobaric tandem mass tag (TMT) labeling, and LC-MS/MS analysis (Supplementary Fig. 7A). From HEK293T cell lysates, the biotin-Ub-PA probe was found to enrich 36 active DUBs compared to the DMSO control (log$_2$ fold change >1, $p$-value ≤ 0.05 in triplicates) (Supplementary Fig. 7B and Supplementary Data 1). Of those, 24 belong to the USP family. The biotin-mISG15$_{CTD}$[WT]-PA probe enriched far fewer proteins, but 3 members of the USP family – USP5, USP14, and USP16 – were highly enriched. This finding is consistent with the previous identification of these DUBs to cross-react with ISG15 in vitro[37]. In contrast, the R153Agb probe showed significantly reduced enrichment of these USP binding partners of ISG15 when there was no detectable USP18 present. We then repeated the ABPP experiment with USP18[WT]-overexpressing HEK293T cell lysates. In this case, we opted to treat with a relatively high probe concentration (5 μM) to ensure sufficient proteome coverage. As expected, USP18 was the protein most highly enriched by the R153Agb probe, followed by USP16 (Fig. 3D). Very minimal or no enrichment was observed for USP5 and USP14, respectively, corroborating the results of recombinant enzyme labeling and immunoblotting of pulldown. Other proteins enriched by the R153Agb probe compared to DMSO (log$_2$ ratio >3) were CPVL, DCAF8, NBAS, and TUBB8. However, their MS intensities were significantly lower than that of USP18 (Supplementary Data 2). Overall, the ABPP data recapitulate the selectivity profile of our USP18 ABP and establish its utility in a quantitative chemoproteomic platform as a chemical tool to selectively report USP18 activity in complex proteomes.

## USP18 inhibitor screening using chemoproteomic platform

With these USP18 ABPs, we sought to perform competitive ABPP to evaluate target engagement and selectivity of DUB inhibitors (Fig. 4A)[41,42,52]. Since USP18 does not efficiently react with Ub, USP18 is excluded in DUB inhibitor screening when using Ub-based probes, despite its biological importance and potential as a drug target. However, we found that several reported DUB inhibitors inhibit USP18 protease activity in ISG15-Rho assays that release the fluorophore rhodamine110 upon USP18-mediated cleavage (Supplementary Fig. 8A, B). Of note, a previously reported USP18 inhibitor contains α,β-unsaturated dienone with two sterically accessible electrophilic β-carbons[53], similar to b-AP15[54] and RA190[55,56], acting as a Michael acceptor to target nucleophiles including the catalytic cysteine of several isopeptidases with a potential chemical liability and pro-miscuity problem. Thus, we chose WP1130 (a known inhibitor of USP5, USP9X, USP14, USP24, and UCH37, Fig. 4B)[57–62], which showed concentration-dependent inhibition of USP18 with an IC$_{50}$ value of

25 μM (Supplementary Fig. 8C), to further characterize in chemoproteomics assays. We confirmed that WP1130 binding competitively prevents the labeling of recombinant USP18 by the ISG15-based covalent probe (Fig. 4C). When USP18 overexpressing HEK293T cell lysates were preincubated with WP1130 for 1 h and treated with each Cy5-Ubl probe (1 μM) for 2 h, we observed a dose-dependent decrease in fluorescent intensity of labeled USP18 and some other DUBs by ISG15-based probes and the Ub-based probe, respectively (Fig. 4D). Compared to other structurally similar compounds (WP1066, AG-490, and G9 in Supplementary Fig. 9A), WP1130 was found to be more potent in competing for probe labeling, consistent with relative potency observed in ISG15-Rho cleavage assays (Supplementary Fig. 9B). When a cocktail of biotin probes (1 μM of Biotin-Ub-PA and 5 μM of Biotin-mISG15$_{CTD}$[R153Agb]-PA) was used to label active DUBs including USP18 for pulldown experiments, immunoblotting analysis of input samples preincubated with WP1130 showed dose-dependent inhibition of USP18 activity (Fig. 4E). These immunoblotting results were supported by quantitative MS analysis with a surprisingly narrow reactivity scope. Among 49 active DUBs detected using a cocktail of probes (log$_2$FC > 1, $p$-value ≤ 0.05 in triplicates, Fig. 4F and Supplementary Data 2), WP1130 binding was found to compete for probe labeling of 4 DUBs – USP22, USP33, USP42, and USP18 – of which USP18 was the most efficiently blocked (Fig. 4G and Supplementary Fig. 10A, B). The reason we were unable to detect other DUBs that WP1130 has been known to target in this experiment might be due to the use of USP18 overexpressing lysates, as we observed WP1130 competitively blocks labeling of USP9X and USP24 in HEK293T cell lysates (Supplementary Fig. 10C and Supplementary Data 3), and the reversible nature of WP1130 binding to targets wherein the outcome of competition assay largely depends on the experimental conditions used. Although not specific, we thus report WP1130 as a small molecule USP18 inhibitor with the potential for optimization. Taken together, our study demonstrates that utilizing USP18 ABPs in combination with Ub-based DUB probes in a chemoproteomics screening platform holds promise as a tractable workflow for global, competitive profiling of DUB inhibitors to perform simultaneous hit identification and selectivity assessment.

## Activity-based profiling of USP18 in lung cancer cell lines

As activity-based protein profiling can determine DUB activities in different pathophysiological conditions[63,64], we chose to profile USP18 activity in lung carcinoma cell lines based on the analysis of expression and gene effect observed from the Broad DepMap database (Supplementary Fig. 11A). A small number of cell lines, A549, H358, H2030, and H1650 lung cancer lines were selected and first evaluated for their USP18 expression. We found that these cell lines express both full-length and truncated USP18 upon IFN-β stimulation (Fig. 5A). Under basal conditions, H358 and H1650 cells appeared to have slightly higher levels of USP18 expressed compared to A549 cells. Relative abundance of USP18 determined by proteomics analysis using label-free, intensity-based absolute quantification (iBAQ) in A549 cells indicated the induced expression of USP18 upon IFN stimulation (Supplementary Fig. 11B). Compared to USP14 and USP5, the abundance of USP18 was found to be lower. Lysates of these lung carcinoma cells were then incubated with Cy5-mISG15$_{CTD}$-PA probe (1 μM, 3 h) and analyzed by SDS-PAGE and in-gel fluorescence scanning (Fig. 5B and Supplementary Fig. 12). We observed fluorescently labeled protein bands around 40-50 kDa in IFN stimulation conditions across cell lines, indicative of USP18–probe conjugates with increased intensity compared to non-IFN stimulation conditions. The fluorescent intensity of the USP18sf (truncated isoform)–probe conjugate band measured by densitometry was slightly higher in H2030 cells. We reason this approach of measuring the differential activity of USP18 combined with expression analysis can help investigate cancer types and states in which USP18 is highly activated and

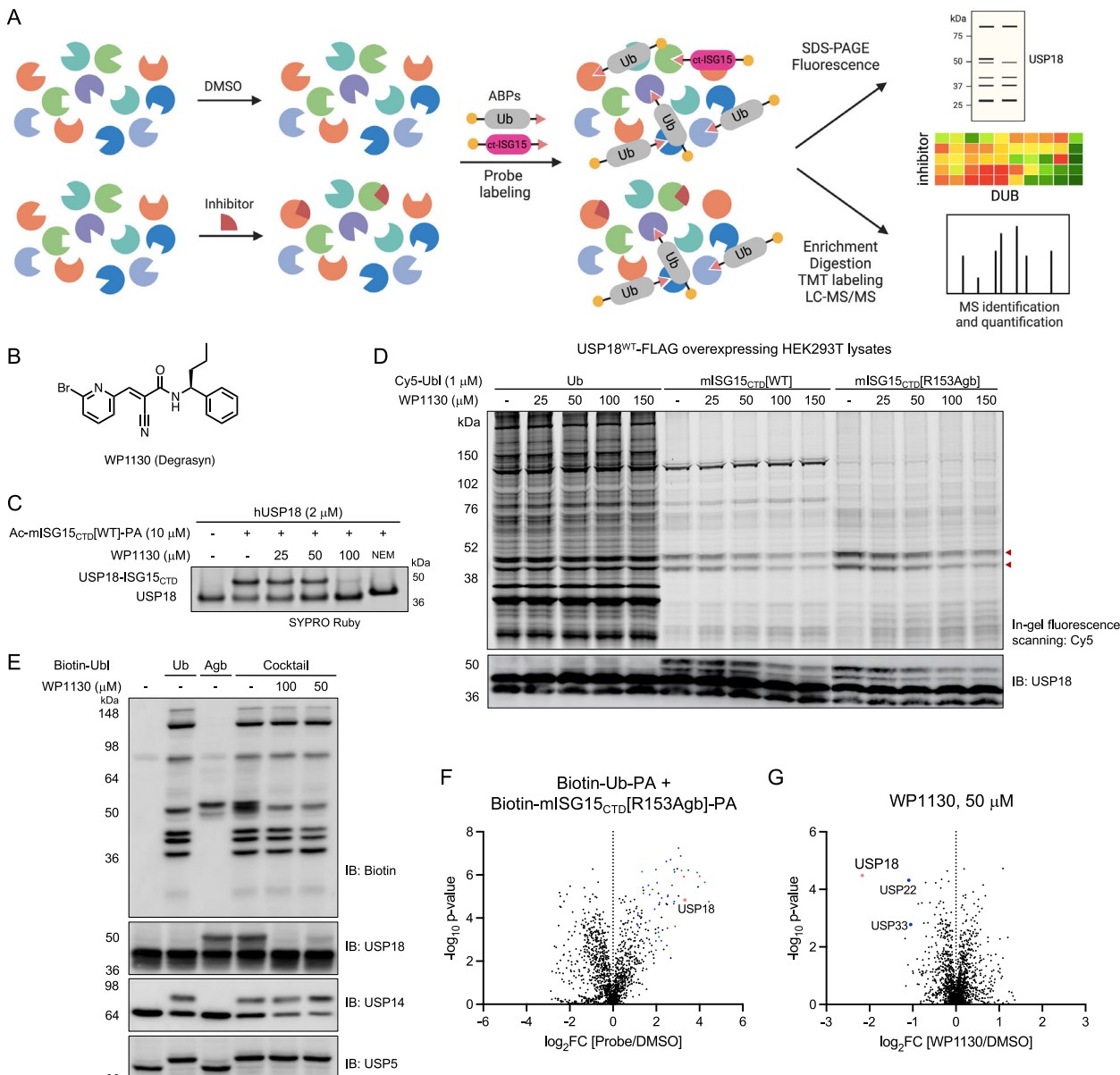

**Fig. 4 | USP18 inhibitor assay by competitive activity-based protein profiling.**
**A** Schematic workflow for quantitative activity-based protein profiling created in BioRender. Yoo, E. (2025) https://BioRender.com/k51r7O1. Lysates were treated with DMSO or compound, followed by treatment with an activity-based DUB probe. Labeled proteins were enriched using biotin-streptavidin pulldown, digested, and analyzed by LC-MS/MS. **B** Chemical structure of WP1130. **C** Recombinant human USP18 (2 μM) was pretreated with WP1130 for 30 min and incubated with Ac-mISG15_CTD[WT]-PA probe (10 μM) for 2 h at RT. Protein samples were analyzed by SDS-PAGE and SYPRO Ruby staining. The result is a representative of two experiments (n = 2 independent replicates). Source data are provided as a Source Data file. **D** USP18^WT-FLAG overexpressing HEK293T cell lysates were preincubated with WP1130 for 1 h followed by labeling with 1 μM of Cy5-Ubl-PA for 2 h at RT. Protein samples were analyzed by SDS-PAGE, in-gel fluorescence scanning for Cy5 signal, and immunoblotting for USP18. The result is a representative of two experiments

(n = 2 independent replicates). Source data are provided as a Source Data file. **E–G** USP18^WT-FLAG overexpressing HEK293T cell lysates were preincubated with WP1130 for 2 h followed by labeling with 1 μM of Biotin-Ub-PA or 5 μM of Biotin-mISG15_CTD[R153Agb]-PA or a cocktail of probes (1 μM of Biotin-Ub-PA + 5 μM of Biotin-mISG15_CTD[R153Agb]-PA) for 3 h at RT. Protein samples were analyzed by SDS-PAGE and immunoblotting (**E**). The result is a representative of two experiments (n = 2 independent replicates). Volcano plots of quantitative proteomic analysis of streptavidin beads pulldowns after pretreating cell lysates with either DMSO (**F**) or 50 μM of WP1130 (**G**) and labeling by a cocktail of probes. Significantly enriched proteins are shown (log_2 ratio >1, p-value ≤ 0.05). USP18 is marked red and other DUBs are colored based on subfamilies (blue: USP, red: UCH, green: OTU, and cyan: JAMM). Data represent mean values (n = 3 independent replicates). p-values were calculated by two-tailed t-test. Source data are provided with this paper (Supplementary Data 2).

provide insights into the cellular context in which USP18 inhibitors can have translational potential.

Since we could detect the USP18 activity in A549 cells upon IFN stimulation, we chose this cell line to perform chemoproteomics analysis using biotin probes. Immunoblotting analysis of input samples incubated with the R153Agb probe indicated labeling of active USP18 with sufficient selectivity over USP5 and USP14 (Supplementary

Fig. 13A). Unexpectedly, labeled USP16 was not detected by the antibody used in this study. Following streptavidin enrichment, MS data demonstrated that USP18 was efficiently pulled down by both WT and R153Agb probes with an enrichment ratio of 2.87 and 2.68, respectively (Fig. 5C, Supplementary Fig. 14, and Supplementary Data 4). The enrichment of USP5 and USP14 by R153Agb probe was reduced relative to the WT probe. However, USP16 was pulled down by both probes at a

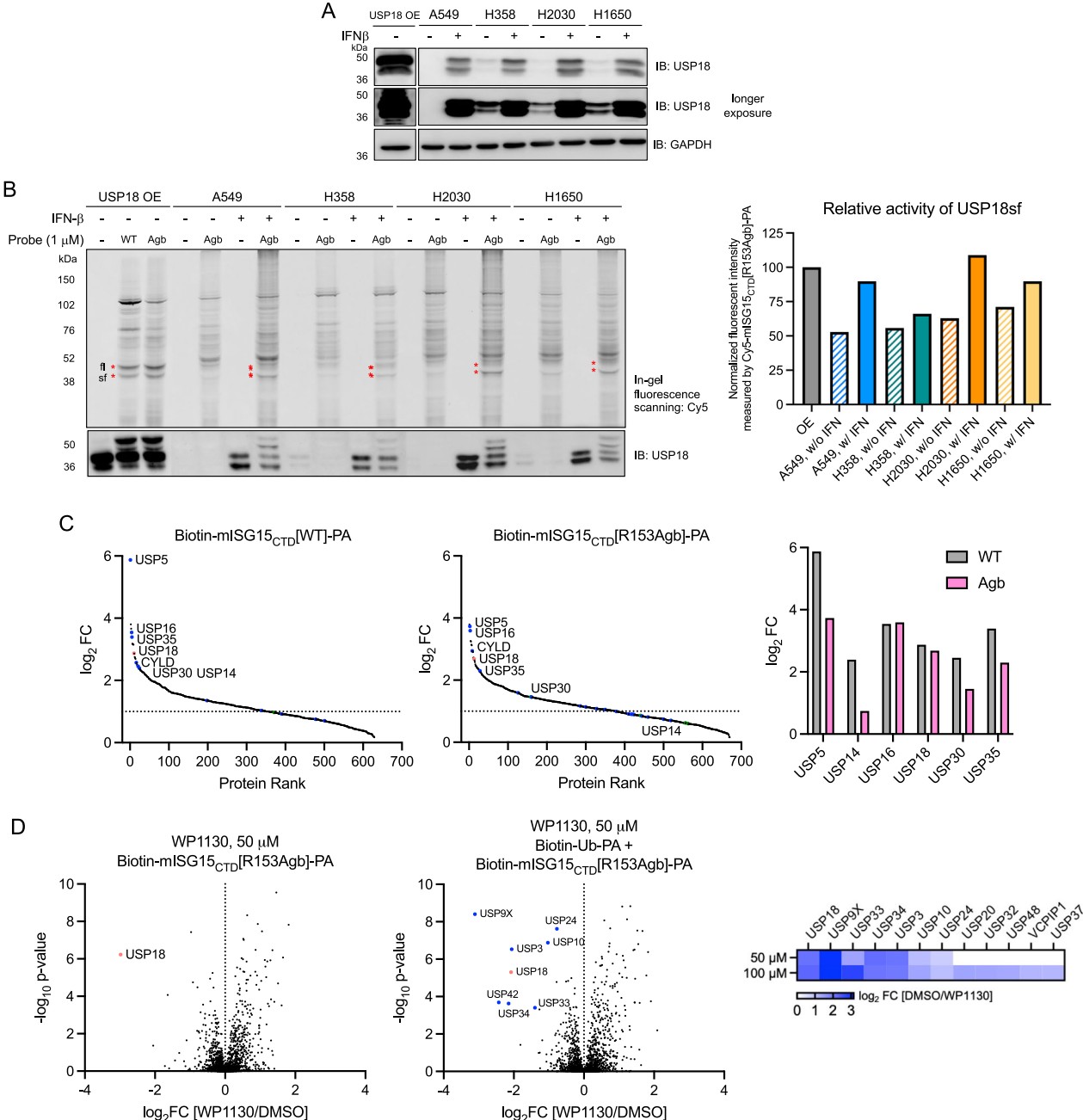

**Fig. 5 | USP18 expression and activity profiles in lung cancer cell lines.**
**A** Constitutive and induced USP18 expression levels upon IFN-β stimulation (50 ng/mL, 48 h) across lung cancer cell lines. Source data are provided as a Source Data file. The result is a representative of three experiments (*n* = 3 independent replicates). **B** Each of the cell lysates were incubated with 1 μM of Cy5-mISG15CTD[R153Agb]-PA probe (3 h, RT). Left, protein samples were analyzed by SDS-PAGE and in-gel fluorescence scanning for Cy5 signal. Red marks indicate the appearance of USP18-probe conjugates. Expression of USP18 was confirmed by western blotting. Right, the normalized intensity of the fluorescent protein band corresponding to USP18sf-probe conjugate to the fluorescent intensity of USP18sf-probe conjugate measured from USP18 overexpressing conditions. The result is a representative of three experiments (*n* = 3 independent replicates). Source data are provided as a Source Data file. **C** Protein rank plots of quantitative proteomic analysis of streptavidin bead pulldowns after labeling of lysates of A549 cells stimulated with IFN-β by 5 μM of either Biotin-mISG15CTD[WT]-PA (left) or Biotin-mISG15CTD[R153Agb]-PA probe (middle), showing significantly enriched proteins (log₂ ratio >1, *p*-value ≤ 0.05). USP18 is marked and other DUBs are

colored based on subfamilies (blue: USP, red: UCH, green: OTU, and cyan: JAMM). Enrichment ratios (log₂ [Probe/DMSO]) of most highly enriched DUBs are compared (right). Data represent mean values (*n* = 3 independent replicates). *p*-values were calculated by two-tailed *t*-test. Source data are provided as a Source Data file and with this paper (Supplementary Data 4). **D** Lysates of A549 cells stimulated with IFN-β were preincubated with 50 μM of WP1130 for 2 h followed by labeling with either 5 μM of Biotin-mISG15CTD[R153Agb]-PA (left) or a cocktail of probes (1 μM of Biotin-Ub-PA + 5 μM of Biotin-mISG15CTD[R153Agb]-PA) (middle). Volcano plots of quantitative proteomic analysis comparing samples treated with WP1130 to DMSO are shown with DUBs inhibited by the compound marked. USP18 is marked and other DUBs are colored based on subfamilies (blue: USP, red: UCH, green: OTU, and cyan: JAMM). Heat map analysis of competition ratios (log₂ [DMSO/WP1130]) comparing samples pretreated with 50 μM or 100 μM of WP1130 followed by labeling with a cocktail of probes indicates several DUBs inhibited by the compound (right). Data represent mean values (*n* = 3 independent replicates). *p*-values were calculated by two-tailed *t*-test. Source data are provided as a Source Data file and with this paper (Supplementary Data 5).

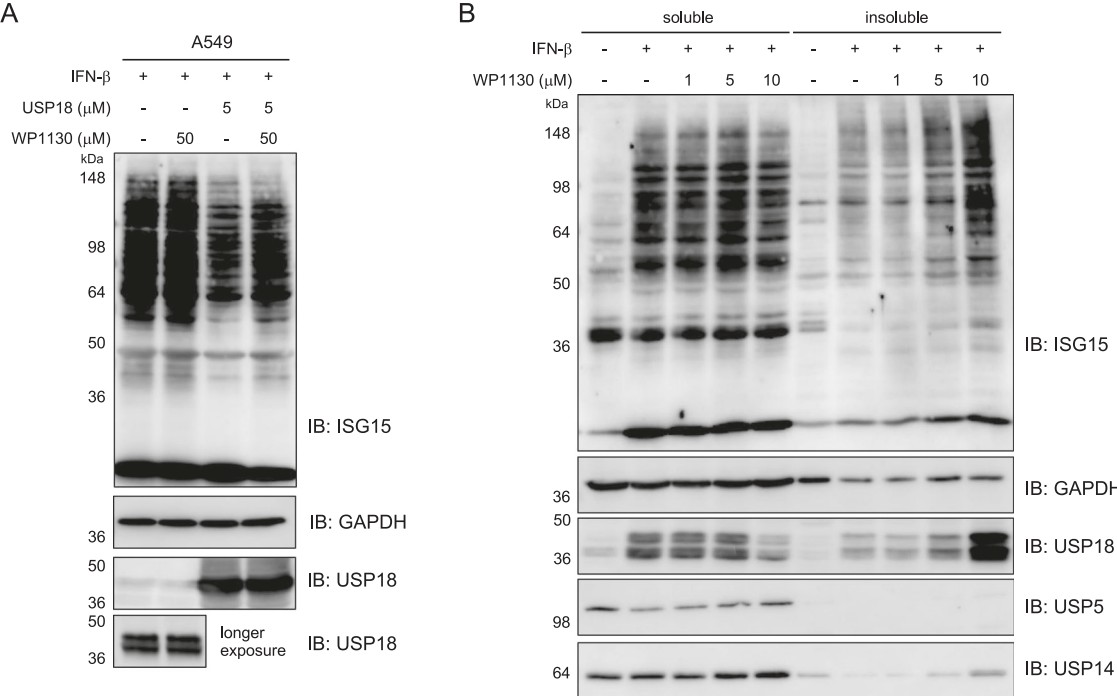

**Fig. 6 | Functional effect of WP1130 on protein ISGylation. A** Lysates of A549 cells stimulated with IFN-β (48 h) were incubated with WP1130 (50 µM) for 2 h in the absence or presence of recombinant human USP18 (5 µM). Protein samples were analyzed by SDS-PAGE and immunoblotted for protein ISGylation. The result is a representative of three experiments ($n$ = 3 independent replicates). Source data are provided as a Source Data file. **B** A549 cells were stimulated with IFN-β (48 h) and incubated with the indicated concentration of WP1130 for 2 h. Lysates were separated into soluble and insoluble fractions and probed for protein ISGylation. The result is a representative of three experiments ($n$ = 3 independent replicates). Source data are provided as a Source Data file.

similar extent, indicating the R153Agb probe reacts with USP16 when its target protein USP18 is expressed at a relatively low level. To avoid the potential complications of cross-reactivity of the probes with USP16, one can design an alternative chemoproteomics strategy where all the Ub-reactive DUBs are blocked with a Ub-based covalent inhibitor prior to labeling with ISG15-based probes. To test this, we performed a labeling assay with fluorescent probes in A549 cell lysates. When we incubated lysates of IFN-stimulated A549 cells with Cy5-mISG15$_{CTD}$[R153Agb]-PA, we could again detect USP18 with decreased labeling of USP14, USP5, or USP16 (estimated based on the molecular weight) compared to WT probe as expected (Supplementary Fig. 15). The labeling of these DUBs that react with both Ub and ISG15 was further diminished when the lysates were preincubated with Ub-PA (1 µM, 1 h) while the labeling of USP18 was maintained. Similar competitive approaches can be applied to the MS-based detection of USP18 using biotin probes.

Additionally, using a cocktail of probes, 1 µM of Biotin-Ub-PA and 5 µM of Biotin-mISG15$_{CTD}$[R153Agb]-PA, which allowed us to detect USP18 along with other 44 DUBs (Supplementary Fig. 14 and Data 4), we carried out competitive ABPP in A549 cell lysates pretreated with WP1130. While USP18 was the only DUB shown to be significantly inhibited by WP1130 among the DUBs detected by the R153Agb probe, we found that several other DUBs in addition to USP18, most notably USP9X, were competed for labeling by a cocktail of probes after WP1130 pretreatment (Fig. 5D, Supplementary Figs. 13B and 16, and Supplementary Data 5). Taken together, these data demonstrate the utility of these USP18 probes in profiling the activity of USP18 in physiologically relevant environments.

After confirming the engagement of USP18 with WP1130 in A549 cell lysates, we examined whether WP1130 has functional and cellular effects. To analyze the effect of WP1130 on protein ISGylation, lysates of A549 cells stimulated with IFN-β were incubated with WP1130 for 2 h. No significant change in ISGylated proteins was observed (Fig. 6A).

When the cell lysates were incubated with recombinant human USP18 (5 µM), a decreased level of ISGylated proteins was detected indicative of deISGylating activity of USP18, which was partially blocked in the presence of WP1130 suggesting that WP1130 inhibits the protein deISGylation. On the other hand, its effect on protein ubiquitylation was not clear (Supplementary Fig. 17A). In A549 cells, WP1130 displayed potent cytotoxic (Supplementary Fig. 18A) and antiproliferative properties (Supplementary Fig. 18B). Next, A549 cells were stimulated with IFN-β for 48 h to induce USP18 expression and cellular ISGylation, and treated with increasing concentrations of WP1130 for 2 h. Cell lysates were then analyzed by immunoblotting. We decided to separate detergent-soluble and insoluble fractions since previous studies have shown that WP1130 treatment leads to rapid accumulation of ubiquitylated proteins in the detergent-insoluble fraction indicative of aggresome formation through DUB inhibition[57,65]. In agreement with that, we observed the accumulation of ubiquitylated, insoluble aggregates of protein in WP1130-treated cells (Supplementary Fig. 17B). Similarly, ISGylated proteins were significantly accumulated in the insoluble fraction upon WP1130 treatment in a dose-dependent manner (Fig. 6B). Interestingly, WP1130 treatment increased USP18 levels detected in the insoluble fraction while a mild decrease was noted in the soluble fraction. It has been reported that ISG15 conjugation marks target substrates for interaction with histone deacetylase 6 (HDAC6) and SQSTM1/p62 toward autophagic clearance[66]. Our study suggests that WP1130 treatment enhances cellular levels of ISGylated proteins that can trigger aggregate formation (aggresomes). Understanding whether these cellular effects of WP1130 observed are direct consequences of USP18 inhibition and if USP18 itself is directly associated with these aggregates warrants additional investigation.

**Assessment of cross-reactivity with USP16**
Although R153Agb mutant probes are less reactive to USP5 and USP14 compared to WT ISG15-based probes, our chemoproteomic data

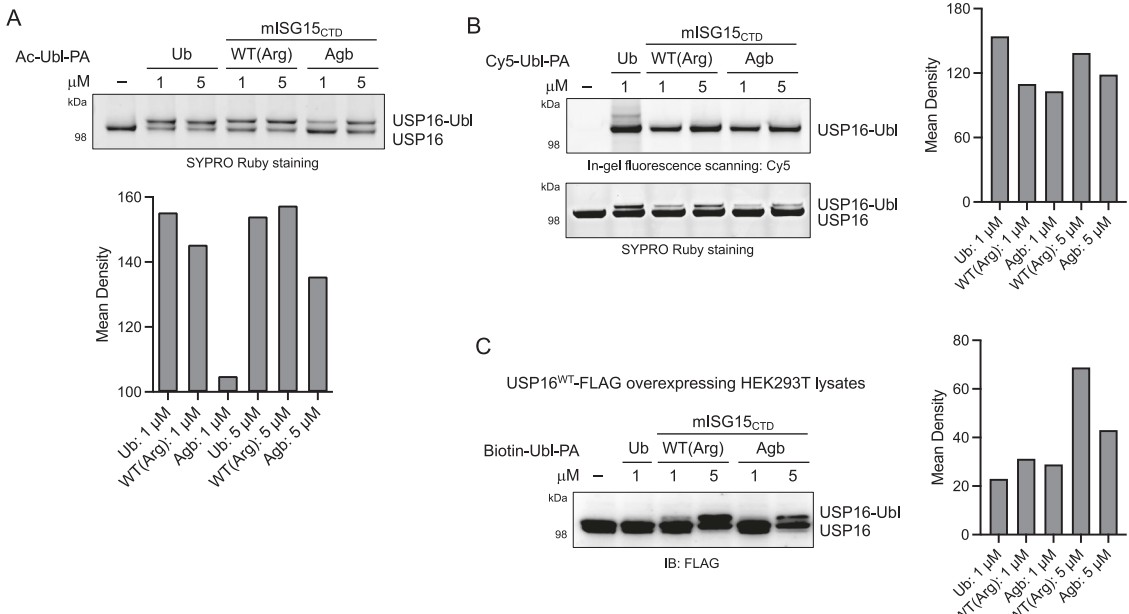

**Fig. 7 | Reactivity with USP16. A** Top, recombinant USP16 (0.45 μM) was incubated with Ac-Ubl-PA probes for 3 h at RT. Protein samples were analyzed by SDS-PAGE and SYPRO Ruby staining. Bottom, mean density of labeled protein band corresponding to the USP16-Ubl probe conjugate. The result is a representative of two experiments ($n = 2$ independent replicates). Source data are provided as a Source Data file. **B** Left, recombinant USP16 (0.25 μM) was incubated with Cy5-Ubl-PA probes (3 h, RT). Protein samples were analyzed by SDS-PAGE, in-gel fluorescence scanning for Cy5, and SYPRO Ruby staining. Right, mean density of labeled protein

band corresponding to the USP16-Ubl probe conjugate. The result is a representative of two experiments ($n = 2$ independent replicates). Source data are provided as a Source Data file. **C** Left, USP16-FLAG overexpressing HEK293T cell lysates were incubated with Biotin-Ubl-PA probes (3 h, RT). Protein samples were analyzed by SDS-PAGE and immunoblotting. Right, mean density of labeled protein band corresponding to the USP16-Ubl probe conjugate. The result is a representative of two experiments ($n = 2$ independent replicates). Source data are provided as a Source Data file.

indicated that they react with USP16 to some extent, especially when USP18 is expressed at a relatively low level. USP16 has been recently identified as an ISG15 cross-reactive DUB that targets pro-ISG15 and ISGylated proteins involved in metabolism[37], in addition to its function in deconjugating Ub and Fubi modification[38,67]. To better understand and characterize the reactivity of our probes toward USP16, we incubated recombinant human USP16 with either Ac-Ubl-PA (Fig. 7A) or Cy5-Ubl-PA (Fig. 7B) and monitored conjugate formation. Both Ub and mISG15$_{CTD}$ probes were found to react and label USP16. Compared to mISG15$_{CTD}$[WT], mISG15$_{CTD}$[R153Agb] mutant showed slightly less intense labeling.

We also synthesized a rhodamine-based fluorogenic substrate incorporating R153Agb mutation of mISG15$_{CTD}$ to measure its cleavage by USP16 in an orthogonal assay. While it was efficiently cleaved by USP18 and generated a fluorescent signal measured at Ex. 485 / Em. 535 nm, its cleavage by USP5 or USP16 was substantially less than the WT substrate (Supplementary Fig. 19).

Since we were unable to detect USP16-probe conjugates in cell lysates using a USP16 antibody (Supplementary Fig. 13), we transiently transfected HEK293T cells with USP16 containing a C-terminal FLAG tag. When lysates were incubated with Biotin-Ubl-PA probes and analyzed by immunoblotting against FLAG, the labeled USP16 by both mISG15$_{CTD}$ probes (5 μM) was detected (Fig. 7C). In addition, to assess the relative reactivity of ISG15-based probes between USP18 and USP16, we incubated an equimolar mixture of recombinant human USP18 and USP16 with Ac-mISG15$_{CTD}$-PA probes and analyzed the extent of labeling of each protein. Both WT and R153Agb mutant probes reacted with USP18 and USP16 to a similar extent (Supplementary Fig. 20). Again, the extent of labeling was comparatively, slightly less with the R153Agb probes at 5 μM.

Together, these results suggest that the R153Agb mutant of mISG15$_{CTD}$ binds and interacts with USP16 but at a comparatively lower level than the WT depending on the concentration used. The ability

and sensitivity of these probes to detect USP18 or USP16 will depend on the relative abundance and activity of these two proteins in the conditions used. To avoid the potential issues of this cross-reactivity, careful considerations need to be taken in designing experiments and interpreting the results.

## Discussion

Despite significant advancement in developing DUB probes, selective chemical tools to study a single DUB are limited. In an effort to develop chemical tools selectively detecting deISGylating activity of USP18 that would allow for the investigation of important (de)ISGylation-dependent biological processes, we identified non-canonical amino acids that can be incorporated into mISG15$_{CTD}$ to preserve reactivity with USP18 while improving selectivity over other ISG15 cross-reactive DUBs such as USP5 and USP14. To understand this gained selectivity, we compared structures of DUB−Ubl complexes within the protease active site. In crystal structures of Ub complexed with USP5 (Supplementary Fig. 21B) and USP14 (Supplementary Fig. 21C), the C-terminal Ub LRGG motif is arranged such that the side chain of R74 forms numerous hydrogen bonds with backbone and side chains of surrounding active site residues (e.g., E427, Q433, Q434 in USP5 and Q196, Q197 in USP14). Shortening the side chain of arginine by a single methylene carbon would disrupt these interactions. On the other hand, unlike Q434 of USP5 or Q197 of USP14, H135 of USP18 is oriented away from the Ubl tail and the side chain of R153 participates in a hydrogen bond interaction in a way that would not be disrupted by substitution with Agb (Supplementary Fig. 21A). Solving the structure of USP18 bound with our Agb mutant probe would provide important information on how to engage major interactions within the active site and determine ISG15 mutants that are susceptible to hydrolysis only by USP18 when conjugated to target substrate protein. Additionally, in this

study, we used a C-terminal domain of the mouse ISG15 sequence as a recognition element for designing chemical probes to profile the protease activity of USP18 because it has shown sufficient binding to both human and mouse USP18, and cross-species compatibility has been reported for ISG15 processing enzymes[68]. Our ongoing work includes the synthesis and characterization of probes using human ISG15 sequence, both a truncated C-terminal and a full length, to determine whether the same mutation incorporation would work with human sequence and understand cross-species interactions of ISG15 with DUBs as sequence variability between species may have a significant impact on molecular interactions. We have already successfully synthesized hISG15$_{CTD}$ using solid-phase peptide chemistry and expect this established protocol will be amenable to incorporating unnatural amino acids.

Using an ABPP platform, our work supports the recent discovery of USP16 as an ISG15 cross-reactive DUB[37,38] and demonstrates an ability to detect USP18 activity in different cell conditions. Although we observed that the reaction of ISG15-based ABPs with USP18 is dependent on its expression level at least in cells tested in this study, since these tools measure the catalytic activity of USP18, the chemical approach presented here can provide insight into contexts where USP18 activity is dysregulated independent of its expression. Moving forward, this strategy should allow us to profile USP18 activity in various cellular and disease states, and eventually elucidate the contribution of its deISGylating activity to the disease phenotype. In particular, our future work will be focused on activity profiling of USP18 in tumor tissue samples. Combined with the Ub-based DUB activity profiling data, the ISG15-based activity profiling of deISGylases and USP18 will help comprehensively investigate and define the activity-related landscape of Ubl proteases and unravel important Ub and Ubl-dependent biological processes in cancer. In addition, adaptable workflow and application in inhibitor screening will provide a road map for developing DUB and USP18 inhibitors. In our initial effort using a small set of compounds, we discovered that a small molecule pan-DUB inhibitor WP1130 inhibits USP18 and enhances the accumulation of ISGylated aggregates. WP1130 displayed cytotoxic and antiproliferative properties in multiple cancer cell lines probably through inhibition of several DUBs. Mechanistic, molecular contribution of USP18 inhibition to these cellular effects, however, remains unclear. WP1130 has also been reported to exhibit potent antiviral effects against norovirus and several RNA viruses[69,70]. As ISGylation pathway plays an important role in antiviral immunity as a defense mechanism, it would be of great interest to investigate USP18 inhibitors in viral infections. All these outstanding questions require more potent and selective ligands. With fluorogenic substrates and ABPs more specific for USP18 in our hands, we have launched a high-throughput biochemical and chemoproteomic screening campaign to identify small molecule USP18 inhibitors and anticipate expanding a chemical toolbox for studying (de)ISGylation pathways.

Since our R153Agb mutant probes bind and interact with USP16 to some extent, our ongoing investigation also includes understanding the molecular interactions between ISG15 and USP16 that can help design more selective probes with increased affinity for USP18 and minimal or no cross-reactivity with USP16. Alternatively, to avoid the potential complications of cross-reactivity, one can deplete the ISG15 cross-reactive DUBs by genetic knockout or immunoprecipitation or design a chemoproteomics method where all the Ub-reactive DUBs are blocked with a Ub-based covalent inhibitor prior to labeling with ISG15-based probes. As we confirmed the feasibility of the latter strategy in a gel-based labeling assay, we anticipate the combination of strategies provided in this study, particularly, utilizing MS-based quantitative proteomics analysis can help investigate functions of USP18 more exclusively in relevant biological settings.

## Methods

### USP18 cloning, protein expression, and purification

Template plasmids for His$_6$-Mm.USP18 (46-368) and His$_6$-Hs.USP18 (16-372) with codon optimization for *Spodoptera frugiperda* (ATUM) were acquired and a TEV protease cleavage site (ENLYFQG) was added by PCR. Using Gateway BP Recombination (ThermoFisher Scientific), entry clones were generated, and both inserts were confirmed by Sanger sequencing. These entry clones were subcloned using Gateway LR Clonase (ThermoFisher Scientific) into pDest-795 (Addgene, 161881), a baculovirus expression vector that contains an N-terminal MBP tag. The expression clones were verified by restriction digestion and these plasmids were transformed into DE95 bacmid strain. Bacmid DNA was isolated by alkaline lysis and transposition of the desired MBP-tev-His$_6$ USP18 ORFS was confirmed by junction PCR.

The two USP18 clones were expressed following the insect cell culture as previously described[71]. The resulting cell pellets were stored at −80 °C. Frozen cell pellets were thawed and resuspended in 100 mL of buffer per 1 L of culture in 20 mM HEPES, pH 7.3, 300 mM NaCl, 1 mM TCEP, and 1:500 (v/v) protease inhibitor cocktail. Lysis was performed using the M-110EH Microfluidizer (Microfluidics Corp., Westwood, MA) for 2 passes at 7000 psi on ice. Lysates were then clarified by ultracentrifugation at 100,000 × g for 30 min at 4 °C and filtered through a Pall 250 ml Autofil 0.45 μm High Flow PES Bottle Top Filter (Thomas Scientific, Swedesboro NJ) and used immediately or stored at −80 °C.

Clarified lysates were thawed, adjusted to 20 mM imidazole, and loaded onto IMAC columns equilibrated in IMAC equilibration buffer (EB) of 20 mM HEPES, pH 7.3, 300 mM NaCl, 1 mM TCEP with 20 mM imidazole. The columns were washed to baseline with EB with 20 mM imidazole and proteins eluted with a 20 column-volume (CV) gradient from 20 to 500 mM imidazole in EB. Elution fractions were analyzed by SDS-PAGE and Coomassie-staining. Positive fractions were pooled, and strep-TEV protease was added at 5% (v/v). The digest proceeded while dialyzing in a minimum buffer volume of 1:20 into EB for 2 h at room temperature followed by new buffer overnight at 4 °C. The digested sample was purified via a second round of IMAC similar to the first round. The sample was loaded to the column equilibrated in EB and washed for 3 CV. Column flow through, wash, and elution bumps were collected as fractions. Columns were eluted with a 3 CV bump to EB with 35 mM imidazole, 3 CV bump in EB with 70 mM imidazole, and 2 CV bump in EB with 500 mM imidazole. After analysis of fractions by SDS-PAGE and Coomassie-staining, fractions containing the target of interest were pooled. The target protein was eluted in the 70 mM imidazole bump and dialyzed using 10k molecular weight cut-off snakeskin into EB. Final samples were assayed for protein concentration, aliquoted in the final buffer (20 mM HEPES, pH 7.3, 300 mM NaCl, and 1 mM TCEP), and frozen.

### Cell culture

HEK293T cells were purchased from American Type Culture Collection (ATCC) and were cultured in Dulbecco's Modified Eagle Medium (DMEM, BTL, 112-013-101) containing 10% (v/v) fetal bovine serum (FBS, VWR International inc., 97068-901) supplemented with 100 units/mL penicillin-streptomycin (Sigma-Aldrich, P4333-100ML) and 2 mM L-glutamine (Gibco, 25030081) at 37 °C in a humidified 5% CO$_2$ incubator. A549 cells were purchased from ATCC (CCL-185) and were cultured in Kaighn's Modification of Ham's F-12 Medium (F-12K, ATCC, 30-2004) containing 10% (v/v) FBS supplemented with 100 units/mL penicillin-streptomycin. H358, H2030 cells were provided by Ji Luo (NCI) and were cultured in Roswell Park Memorial Institute 1640 media (RPMI-1640, BTL, 112-024-101) containing 10% (v/v) FBS supplemented with 100 units/mL penicillin-streptomycin and 2 mM L-glutamine. H1650 cells were purchased from ATCC (CRL-5883) and were cultured in RPMI-1640 containing 10% (v/v) FBS supplemented with 100 units/ mL penicillin-streptomycin and 2 mM L-glutamine.

## Stimulation of cells

Cells were treated with 50 ng/mL of recombinant human IFN-β (PeproTech, 300-02BC) for the indicated times (48 h).

## Transfection

pcDNA3.1-USP18 and pcDNA3.1-USP16 plasmids were purchased from GenScript. The plasmid was transformed into DH5α cells (Invitrogen, 18258012) The following day, a single transformed colony was used to inoculate 50 ml of LB medium (LB broth Lennox, Sigma-Aldrich, L3022-250g) containing 100 μg/ml of ampicillin (Sigma-Aldrich, A9518) and was incubated at 37 °C overnight with agitation (250 rpm). A Midi prep (Qiagen, 12943) kit was used to isolate the plasmid for further experiments. HEK293T cells were grown in completed DMEM and maintained at 37 °C with 5% CO$_2$. Each 100 mm cell culture dish was contained with 10 ml of complete DMEM media and transfected with 10 μg of overexpression plasmid with 30 μL Lipofectamine LTX (Invitrogen, 15338100) in 1 ml of Opti-MEM (Gibco, 31985070). After 48 h, cells were collected in PBS with 2 times of wash.

## Cell viability assay

A549 cells were seeded in 96-well plates overnight. The cells were then treated with DMSO vehicle control or serial doses of WP1130 and incubated at 37 °C for 6 h. Cell viability assay was performed using CellTiter-Blue® Cell Viability Assay reagent (Promega, G8081) according to the manufacturer's protocol. Fluorescent signals (Ex. 560(20) / Em. 590(10)) were measured using the BioTek Cytation 5 (Agilent).

## Cell proliferation assay

A549 cells were seeded in 96-well plates. On the next day, cells were treated with DMSO vehicle control or serial doses of WP1130 and incubated at 37 °C for 48 h. Cell Proliferation assay was performed using CellTiter-Blue® Cell Viability Assay reagent (Promega, G8081) according to the manufacturer's protocol at 0 h and 48 h time point. Fluorescent signals (Ex. 560(20) / Em. 590(10)) were measured using the BioTek Cytation 5 (Agilent).

## ISG15-Rh110 cleavage assay

Recombinant DUBs at optimal concentrations (mUSP18, 2.5 nM or 10 nM; USP5, 500 nM; USP16, 10 nM or 25 nM) were incubated in assay buffer, 50 mM Tris (pH 7.5), 100 mM NaCl, 1 mM TECP, 0.01% (w/v) BSA, 0.01% (v/v) Triton X-100, containing DUB inhibitor (indicated concentration), vehicle alone (DMSO) in a 20 μL reaction volume for 30 to 60 min at 37 °C. The reaction was initiated by the addition of fluorogenic substrate Ac-mISG15$_{CTD}$-Rh110 (1.5 μM or 2 μM), and the release of rhodamine fluorescence was recorded at Ex. 485 nm / Em. 535 nm for 1 h at room temperature using a microplate reader.

## Immunoblotting for protein ISGylation

In situ. A549 cells were seeded in 6-well plate overnight and then treated with IFN-β (50 ng/ml) at 37 °C for 48 h. Subsequently, DMSO vehicle control or serial doses of WP1130 were added and incubated at 37 °C for 2 h. Cells were then washed with PBS twice and lysed by RIPA Lysis and Extraction Buffer (Thermo Scientific™, 89900) for the soluble fraction. After incubation with RIPA Lysis buffer, cell lysates were centrifuged (30,000 × g, 30 min) to separate the pellet. The insoluble fraction was treated with 2x SDS sample buffer and 5 cycles of freeze and thaw. Total protein amount was normalized by Pierce™ BCA Protein Assay Kits (Thermo Scientific™, 23227). The samples were analyzed by SDS-PAGE and transferred onto a nitrocellulose membrane (Bio-Rad). The membranes were blocked with 5% (w/v) skim milk (Cell Signaling Technology, 9999S) in TBST(0.1% (v/v) of Tween 20 in TBS buffer) for 1 h, washed 3 times with TBST and then incubated with 1:1000 ratio of USP18 (Cell Signal Technology, 4813S), USP5 (Bethyl, A301-542A), USP14 (Cell Signal Technology, 11931S), ISG15 (Cell Signal Technology, 2743S), Ubiquitin (Cell Signal Technology, 58395S), and

GAPDH (Cell Signal Technology, 5174S) primary antibody in 2% BSA/TBST for overnight at 4 °C. After the overnight incubation, membranes were washed 3 times with TBST and probed with a 1:2000 ratio of horseradish peroxidase-conjugated secondary antibody (Cell Signaling, 7074) or 1:4000 ratio (Sigma-Aldrich, 12-348) in 2% BSA/TBST for 1 h at room temperature, followed by ECL detection (SignalFire ECL Reagent, Cell Signaling or SuperSignal™ ELISA Femto Substrate, Thermo Scientific™). The protein bands were observed using ImageQuant 800 imaging system (Cytiva).

In vitro. IFN-β-stimulated A549 cells were lysed in 50 mM Tris (pH 7.4), 150 mM NaCl, 2 mM EDTA, 0.5% NP-40, and 1 mM TCEP. The final concentration of 5 μM recombinant human USP18 protein was spiked into the 2 mg/ml of cell lysates. The lysates were incubated with 50 μM of WP1130 for 2 h at room temperature. The reaction was stopped by the addition of reducing Laemmli SDS 4X sample buffer (Thermo Scientific™, J60015.AC) and boiling. The samples were analyzed by SDS-PAGE and western blot.

## DUB labeling assay

Recombinant proteins were diluted in 25 mM Tris (pH 7.5), 150 mM NaCl, and 10 mM DTT. Purified Ac-mISG15$_{CTD}$-PA probes were diluted in 50 mM Tris (pH 7.5), 50 mM NaCl, and 5 mM DTT. USP18 (2 μM), USP5 (1 μM), or USP16 (0.45 μM or 0.25 μM) were reacted with 10 μM of probe for 3 h at room temperature. Reaction was stopped by addition of sample buffer and labeling was visualized by SDS-PAGE using NuPAGE 4–12% Bis-Tris gel with MES (for USP18) or MOPS (for USP15 and USP16) running buffer and SYPRO Ruby staining.

## Activity-based protein profiling (ABPP) in lysates

**Gel-based ABPP.** Using HEK293T cells transfected with plasmids genetically encoding full-length USP18$^{WT}$ and the catalytic mutant USP18$^{C64/65A}$, lysate was prepared 48 h after transfection in 50 mM Tris-HCl, 5 mM MgCl$_2$, 250 mM sucrose, and 1 mM DTT. 40 μg of lysate (20 μL, 2 mg/mL) was treated with Cy5-Ubl-PA probe for 3 h at room temperature. Reaction was stopped by addition of reducing sample buffer and boiling. The samples were analyzed by SDS-PAGE and in-gel fluorescence scanning for Cy5 signal.

A549, H358, H1650, H2030 cells were prepared with or without IFN-β stimulation. Cell lysates were prepared in 50 mM Tris-HCl, 5 mM MgCl$_2$, 250 mM sucrose, and 1 mM DTT. 40 μg of total protein was treated with Cy5-Ubl-PA, WT, and Agb probe for 3 h at room temperature. Reaction was stopped by addition of reducing sample buffer and boiling. The samples were analyzed by SDS-PAGE and in-gel fluorescence scanning for Cy5 signal.

**Competitive gel-based ABPP.** 40 μg of lysate (20 μL, 2 mg/mL) was treated with either Ac-mISG15$_{CTD}$[WT]-PA, Ac-mISG15$_{CTD}$[R153Agb]-PA or Ub-PA and allowed to incubate for 1 h at room temperature. Samples were then chased with 1 μM Cy5-mISG15$_{CTD}$[WT]-PA for 2 h at room temperature, and the reaction was stopped by addition of reducing sample buffer and boiling. The samples were analyzed by SDS-PAGE and in-gel fluorescence scanning for Cy5 signal.

**Western blotting.** Cell lysates were mixed with reducing Laemmli SDS sample 4x buffer (Thermo Scientific™, J60015.AC) and analyzed by SDS-PAGE. Samples were then transferred onto a nitrocellulose membrane (Bio-Rad). The membranes were blocked with 5% (w/v) skim milk (Cell Signaling Technology, 9999S) in TBST(0.1% (v/v) of Tween 20 in TBS buffer) for 1 h, washed 3 times with TBST and then incubated with 1:1000 ratio of USP18 (Cell Signal Technology, 4813S), USP5 (Bethyl, A301-542A), USP14 (Cell Signal Technology, 11931S), USP16 (Bio-Rad, VPA00705), and Biotin(Cell Signal Technology, 5597S) primary antibody in 2% BSA/TBST for overnight at 4 °C. After the overnight incubation, membranes were washed 3 times with TBST and probed with a 1:2000 ratio of horseradish peroxidase-conjugated

secondary antibody (Cell Signaling, 7074) or 1:4000 ratio (Sigma-Aldrich, 12-348) in 2% BSA/TBST for 1 h at room temperature, followed by ECL detection (SignalFire ECL Reagent, Cell Signaling or Super-Signal™ ELISA Femto Substrate, Thermo Scientific™). The protein bands were observed using ImageQuant 800 imaging system (Cytiva).

**Quantitative MS-based ABPP.** HEK293T or A549 cells were lysed (50 mM Tris pH 8.0, 150 mM NaCl, 5 mM MgCl$_2$, 0.5 mM EDTA, 0.5% NP-40, 10% glycerol, 1 mM TCEP, protease inhibitor (cOmplete™ Mini Protease Inhibitor Cocktail, Roche)), clarified by centrifugation, and then diluted to a concentration of 2 mg/mL. 500 µL aliquots were incubated with 50 or 100 µM of the WP1130 or DMSO for 2 h at room temperature. Subsequently, Biotin-Ubl-PA probe was added and incubated for 3 h at room temperature. To quench the reaction, 500 µL of 0.2% SDS, 0.5% NP-40 in PBS were added to cell lysate (final: 0.1% SDS, 0.25% NP-40). 100 µL of high-capacity streptavidin agarose resin slurry (ThermoFisher Scientific) was added to low-binding micro-centrifuge tubes (Sorenson) and washed (3 × PBS) and 200 µL of 0.2% NP-40/PBS added before mixing with cell lysate. Protein samples were added to the streptavidin beads and incubated at room temperature for 1 h with end-to-end rotation. After incubation, beads were washed (4× 0.2% NP-40/PBS, 4x PBS, 4x ddH$_2$O, 1x HEPES) by centrifugation (500 × $g$, 1 min, 4 °C) with Micro Bio-Spin™ Chromatography Columns (Bio-rad). After the final wash, the supernatant was removed, 50 µL of 100 mM of HEPES pH 7.4 was added, and the beads were flash-frozen in liquid N$_2$ and stored at −80 °C.

**Identification of probe-labeled proteins by mass spectrometry.** The frozen streptavidin beads were thawed at room temperature and each were added with 200 µL of 1:1:1:1 100 mM HEPES/lysis buffer/reducing solution/alkylating solution from the EasyPep™ MS Sample Prep Kit (Thermo, A40006) containing 3 µg Trypsin/LysC protease mix (Thermo, A40007). The samples were incubated overnight at 37 °C with shaking and 200 µL aliquot from each sample was obtained. TMTpro 18-plex reagents (125 µg, Thermo, A52045) were dissolved in 20 µL acetonitrile (ACN) and added into each sample, followed by incubation at 25 °C for 1 h. Samples were quenched with 50 µL of 5% hydroxylamine + 20% formic acid (FA) for 15 min at room temperature. The samples were pooled and cleaned up using EasyPep Mini columns (Thermo, A40006) by following the manufacturer's protocol. The eluted peptides were dried under vacuum in a speed-vac and stored at −20 °C.

The dried peptides were resuspended in 50 µL of 0.1% FA and loaded 5 µL twice onto a Dionex U3000 RSLC in front of a Orbitrap Eclipse (Thermo) equipped with an EasySpray ion source Solvent A consisted of 0.1% FA in water and Solvent B consisted of 0.1% FA in 80% ACN. Loading pump consisted of Solvent A and was operated at 7 µL/min for the first 6 min of the run then dropped to 2 µL/min when the valve was switched to bring the trap column (Acclaim™ PepMap™ 100 C18 HPLC Column, 3 µm, 75 µm I.D., 2 cm, PN 164535) in-line with the analytical column (EasySpray C18 HPLC Column, 2 µm, 75 µm I.D., 25 cm, PN ES902). The gradient pump was operated at a flow rate of 300 nL/min and each run used a linear LC gradient of 5–7%B for 1 min, 7–30% B for 134 min, 30–50% B for 35 min, 50–95% B for 4 min, holding at 95% B for 7 min, then re-equilibration of analytical column at 5% B for 17 min. All MS injections employed the TopSpeed method with four FAIMS compensation voltages (CVs) and a 0.75 s cycle time for each CV (3 s cycle time total) that consisted of the following: Spray voltage was 2200 V and ion transfer temperature of 300 °C. MS1 scans were acquired in the Orbitrap with resolution of 120,000, AGC of 4e5 ions, and max injection time of 50 ms, mass range of 375–1600 $m/z$; MS2 scans were acquired in the Orbitrap using TurboTMT method with resolution of 15,000, AGC of 1.25e5, max injection time of 22 ms, HCD energy of 38%, isolation width of 0.4 Da, intensity threshold of 2.5e4 and charges 2–6 for MS2 selection. Advanced Peak

Determination, Monoisotopic Precursor selection (MIPS), and EASY-IC for internal calibration were enabled and dynamic exclusion was set to a count of 1 for 15 s. The only difference in the methods was the CVs used: one method used CVs of −45, −55, −65, −75 and the second used CVs of −50, −60, −70, −80.

**Database search and data post-processing.** Both injections for each sample were pooled together as fractions and all MS files were searched with Proteome Discoverer 2.4 using the Sequest node. Data was searched against the Uniprot Human database from Feb 2020 using a full tryptic digest, 2 max missed cleavages, minimum peptide length of 6 amino acids and maximum peptide length of 40 amino acids, an MS1 mass tolerance of 10 ppm, MS2 mass tolerance of 0.02 Da, variable oxidation on methionine (+15.995 Da) and fixed modifications of carbamidomethyl on cysteine (+57.021), TMTpro (+304.207) on lysine and peptide N-terminus. Percolator was used for FDR analysis and TMTpro reporter ions were quantified using the Reporter Ion Quantifier node and normalized on total peptide intensity of each channel. TMTpro channel assignment for conditions can be found in supplementary data. Data represent mean values ($n = 3$ independent replicates). $p$-values were calculated by two-tailed $t$-test. TMT-MS data have been deposited at MassIVE with accession number MSV000094406.

**Combinatorial substrate library synthesis**
A combinatorial tetrapeptide fluorogenic substrate library was synthesized on a solid support according to published protocols[43,46]. The library consisted of two tetrapeptide sub-libraries. Each of the sub-libraries was synthesized separately, and the general synthetic procedure is described for the P3 sub-library as an example. In the first step, Fmoc-ACC-OH (25 mmol, 2.5 eq.) was attached to the Rink amide resin (13.5 g) using coupling reagents: HOBt (25 mmol, 2.5 eq.) and DICI (25 mmol, 2.5 eq.) in DMF. After 24 h, the Fmoc protecting group was removed with 20% piperidine in DMF. In the next step, Fmoc-Gly-OH (25 mmol, 2.5 eq.) was coupled using HATU (25 mmol, 2.5 eq.) and 2,4,6-collidine (25 mmol, 2.5 eq.) in DMF. Then, Fmoc-Gly-OH (25 mmol, 2.5 eq.) was attached to the H$_2$N-Gly-ACC-resin using HOBt and DICI (25 mmol, 2.5 eq.) as coupling reagents. After glycine coupling, the Fmoc group was removed (20% piperidine in DMF), and the resin was washed with DCM and MeOH and dried over P$_2$O$_5$. Then, the dried resin was divided into 138 portions. To each portion of the H$_2$N-Gly-Gly-ACC-resin, natural or unnatural amino acids were attached, and the Fmoc protecting group was removed (20% piperidine in DMF). To provide an equimolar substitution of each natural amino acid in the P4 position, an isokinetic mixture of Fmoc-protected amino acids was utilized. The last two steps of P3 sub-library synthesis included N-terminal acetylation (solution of AcOH, HBTU and DIPEA) and cleavage of peptides from the resin using a TFA:H$_2$O:TIPS (95:2.5:2.5, % v/v/v) mixture. Finally, the sub-library was precipitated in Et$_2$O, dissolved in a mixture of acetonitrile and water, lyophilized, and dissolved in biochemical grade DMSO at a concentration of 20 mM. The obtained sub-library was used for kinetic studies without further purification. The P4 sub-library was synthesized in the same manner.

**Substrate library screening**
All screenings were performed using a spectrofluorometer (Molecular Devices Spectramax Gemini XPS) in 96-well plates (Corning). The release of ACC was monitored continuously for 40 min at the appropriate wavelength (Ex = 355 nm, Em = 460 nm). For the assay, 0.5 µL of the substrate in DMSO was used with 49.5 µL of the enzyme. The enzyme was incubated in assay buffer (50 mM Tris, 100 mM NaCl, 1 mM TCEP, 0.01% BSA, 0.01% Triton X-100, pH 7.5) for 30 min at 37 °C before addition to the substrates on a plate. The final substrate concentration in each well during the assays was 200 µM for combinatorial P3 and P4 sub-libraries. The enzyme concentration was 10 µM for mUSP18. The linear part of each progress curve was used to determine

the substrate hydrolysis rate. Substrate specificity profiles were established by setting the highest value of relative fluorescence unit per second (RFU/s) from each library position as 100% and adjusting other values accordingly.

### Surface plasmon resonance (SPR) analysis

Surface plasmon resonance binding analysis was performed on a BIAcore 3000 (GE Healthcare) instrument. Mouse and human USP18 were immobilized via the N-terminal His$_6$ tag on NTA chip (Cytiva). Protein binding analysis was performed at 25 °C in 50 mM HEPES, 150 mM NaCl, 0.05 mM EDTA, 0.5 mM DTT and 0.01% P20, pH 7.5, with a flow rate of 20 μL/min. Stock solutions of ubiquitin, ISG15 and mouse ISG15 variants were diluted in running buffer. Binding traces were analyzed with BIAevaluation 4.0 software (GE Healthcare) and fitted with a Langmuir 1:1 binding model including a drift of the baseline.

### Preparation of ISG15-based probes

**Ac-hISG15CTD-VS.** The C-terminal domain of human ISG15 was synthesized using solid-phase peptide synthesis (SPPS) with Fmoc-Gly Tentagel R Trt resin (RAPP Polymere 0.17 mmol/g) to yield the sequence Ac-SILVRNNKGRSSTYEVRLTQTVAHLKQQVSGLEGVQDDLFWLTFEGKPLE DQLPLGEYGLK PLSTVFMNLRLRG (wherein the underlined portions denote positions where pseudoproline dipeptides were used). All sequences were synthesized at a 25 μmol scale in *N,N*-dimethylformamide (DMF) with HCTU (4 eq., 100 μmol, 41 mg) and diisopropylethylamine (8 eq., 200 μmol, 35 μL) using Tetras synthesizer (Thuramed) with Fmoc-based SPPS, and reaction progress was periodically monitored by microcleavage. Upon completion, 12.5 μmol of the protected peptide was cleaved from the resin using 20% (v/v) of 1,1,1,3,3,3-hexa-fluoroisopropanol in dichloromethane (DCM) and the cleavage mixture was removed by evaporation, after which the cleaved peptide was coevaporated with 1,2-dichloroethane. After drying under vacuum, the resulting peptide was resuspended in 3 mL dry DCM and stirred overnight with PyBOP (4 eq., 50 μmol, 26 mg), triethylamine (4 eq., 50 μmol, 7 μL), and Gly-vinyl sulfone (10 eq., 125 μmol, 17 μg). All solvents were evaporated, and the peptide mixture was then fully deprotected by stirring in TFA/H$_2$O/TIPS/phenol (90:5:2.5:2.5 (v/v/v/v)) for 3.5 h. The crude, deprotected peptide was precipitated from the cleavage mixture through addition of chilled ether:*n*-heptane = 3:1 solution (v/v) and spun down. The pellet was allowed to air dry, and then resuspended in H$_2$O:ACN:AcOH = 65:25:10 (v/v/v) and lyophilized. The crude product was then purified by RP-HPLC on a Teledyne ACCQ Prep HP 150 with a C18 column (Redisep). The mobile phases were; A, 0.1% TFA in H$_2$O; B, 0.1% TFA in ACN. At a flow rate of 20 mL/min, a gradient of 20–80% B over 35 min was used. Fractions containing the pure product were combined and lyophilized.

**Ac-mISG15CTD-PA.** The C-terminal domain of mouse ISG15 was synthesized using solid-phase peptide synthesis (SPPS) to yield the sequence Ac-LSILVRNERGHSNIYE VFLTQTVDTLKKKVS QREQVHEDQF WLSFEGRPMEDKELLGEYGLKPQSTVIKHLRLRG (wherein the underlined portions denote positions where pseudoproline dipeptides were used, and the bolded residue in position 144 indicates a Cys→Ser change from the native mISG15 sequence)[47]. For the R153Agb and R153F(guan) mutant sequences, non-canonical amino acids were manually coupled to pre-loaded Fmoc-Gly Tentagel R Trt resin (RAPP Polymere 0.17 mmol/g) prior to automated synthesis. All sequences were synthesized at a 25 μmol scale in *N*-methylpyrrolidone (NMP) with PyBOP (4 eq., 100 μmol, 52 mg) and diisopropylethylamine (8 eq., 200 μmol, 35 μL) using Tetras synthesizer (Thuramed) with Fmoc-based SPPS, and reaction progress was periodically monitored by microcleavage. Upon completion, 12.5 μmol of the protected peptide was cleaved from the resin using 20% (v/v) of 1,1,1,3,3,3-hexa-fluoroisopropanol in dichloromethane (DCM) and the cleavage mixture was removed by evaporation, after which the cleaved peptide was

coevaporated with 1,2-dichloroethane. After drying under vacuum, the resulting peptide was resuspended in 3 mL dry DCM and stirred overnight with PyBOP (4 eq., 50 μmol, 26 mg), triethylamine (4 eq., 50 μmol, 7 μL), and propargylamine (10 eq., 125 μmol, 8 μL). All solvents were evaporated, and the peptide mixture was then fully deprotected by stirring in TFA/H$_2$O/TIPS/phenol (90:5:2.5:2.5 (v/v/v/v)) for 3.5 h. The crude, deprotected peptide was precipitated from the cleavage mixture through addition of chilled ether:*n*-heptane = 3:1 solution (v/v) and spun down. The pellet was allowed to air dry, and then resuspended in H$_2$O:ACN:AcOH = 65:25:10 (v/v/v) and lyophilized. The crude product was then purified by RP-HPLC on a Teledyne ACCQ Prep HP 150 with a C18 column (Redisep). The mobile phases were; A, 0.1% TFA in H$_2$O; B, 0.1% TFA in ACN. At a flow rate of 20 mL/min, a gradient of 20–80% B over 30 min was used. Fractions containing the pure product were combined and lyophilized.

**Biotin-mISG15CTD-PA.** The protocol for SPPS was followed exactly as specified above. 12.5 μmol of resin was swelled in NMP for 30 min prior to Fmoc-deprotection in piperidine/NMP (20% (v/v), 30 min). The resin was then allowed to shake overnight with biotin-X-NHS (4 eq., 50 μmol, 23 mg) and diisopropylethylamine (16 eq., 200 μmol, 35 μL) in 3 mL NMP. Biotinylated polypeptide was then cleaved from the resin using 1,1,1,3,3,3-hexafluoroisopropanol in DCM (20% (v/v), 2x 30 min), and all steps for C-terminal propargylation were followed as described above.

**Cy5-mISG15CTD-PA.** The protocol for SPPS was followed exactly as specified above. 12.5 μmol of resin was swelled in NMP for 30 min prior to Fmoc-deprotection in piperidine/NMP (20% (v/v), 30 min). The resin was then washed with NMP and DMF, successively. The resin was then allowed to shake overnight in the dark with Cy5-NHS ester (1 eq., 12.5 μmol, 8 mg) and diisopropylethylamine (16 eq., 200 μmol, 35 μL) in 3 mL DMF. Cy5-coupled polypeptide was then cleaved from the resin using 1,1,1,3,3,3-hexafluoroisopropanol in DCM (20% (v/v), 2x 30 min), and all steps for C-terminal propargylation were followed as described above.

**Ac-mISG15CTD-OH.** The protocol for SPPS was followed exactly as specified above. The completed peptide was then cleaved from the resin and fully deprotected simultaneously, through treatment with TFA:H$_2$O:TIPS:phenol = 90:5:2.5:2.5 (v/v/v/v) for 3.5 h. All precipitation and purification steps were followed as specified above.

**Ac-mISG15CTD-Rh110.** The protocol for SPPS was followed exactly as specified above. Upon completion, 25 μmol of the protected peptide was cleaved from the resin using 20% (v/v) of 1,1,1,3,3,3-hexa-fluoroisopropanol in dichloromethane (DCM) and the cleavage mixture was removed by evaporation, after which the cleaved peptide was coevaporated with 1,2-dichloroethane. After drying under vacuum, the resulting peptide was resuspended in 4 mL dry DCM and stirred overnight in the dark with PyBOP (5 eq., 125 μmol, 65 mg), triethylamine (20 eq., 500 μmol, 150 μL), and BisGly-Rh110 (10 eq., 250 μmol, 112 mg). All solvents were evaporated, and the peptide mixture was then fully deprotected by stirring in TFA/H$_2$O/TIPS/phenol (90:5:2.5:2.5 (v/v/v/v)) for 3.5 h. All precipitation and purification steps were followed as specified above.

### Reporting summary

Further information on research design is available in the Nature Portfolio Reporting Summary linked to this article.

## Data availability

The data supporting the findings of this study are available within the article and Supplementary Information. TMT-MS data have been deposited at MassIVE with accession number MSV000094406. Source data of all uncropped gels and blots are provided in the Source Data

file. The protein structures were retrieved from the Protein Data Bank with the accession codes: 5CHV; 3IHP and 2AYO. Source data are provided with this paper.

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

## Acknowledgements

The cloning, expression, and purification of mouse and human USP18 were done at the Protein Expression Laboratory, Frederick National Laboratory for Cancer Research, and the authors are grateful to Dr. Jane Jones's team for their assistance. The authors thank Dr. Ji Luo's group (Laboratory of Cancer Biology and Genetics, CCR, NCI) for providing two lung cancer cell lines used in this study (H358 and H2030). The authors gratefully thank the staff members at the CCR-Frederick Biophysics Resource for their assistance, technical consultation, and instrument maintenance. This work was supported by the Intramural Research Program of the NIH, National Cancer Institute, Center for Cancer Research (ZIABC011963, E.Y.), National Science Center grant 2015/17/N/ST5/03072 (Preludium 9) in Poland (W.R.) and the "TEAM/2017-4/32" project, which is carried out within the TEAM program of the Foundation for Polish Science, cofinanced by the European Union under the European Regional Development Fund (M.D.).

## Author contributions

G.J.D. and A.O.O. synthesized chemical probes and conducted DUB cleavage and labeling assays; Y.J. conducted biochemical and cellular experiments; W.R. performed HyCoSuL screening; R.H. and K.F.S conducted the proteomics studies and performed data analysis; G.J.D., H.K., and M.Y performed biophysical assays; T.A., M.D., and E.Y. assisted in initial method development and project supervision; E.Y. designed and supervised the project and, with contributions from all authors, wrote the manuscript.

## Funding

## Competing interests

The authors declare no competing interests.
