## [Transparent Peer Review file · Nature Communications]

Chemical tools to define and manipulate interferon-inducible Ubl protease USP18

Corresponding Author: Dr Euna Yoo

Version 0:

Reviewer comments:

Reviewer #1

(Remarks to the Author)

Here, Davis et al describe a new ISG15 activity based probe (ABP) containing unnatural amino acids (R153AGB). Authors propose that this probe exhibits enhanced selectivity towards USP18 as compared to a wt ABP.

This probe can be a valuable tool, but due to its artificial nature requires substantial testing. Authors have addressed this to a certain extent but some important issues need to be addressed:

1. One of the most critical experiments to determine the use in cells is the proteomic analysis of streptavidin pull downs from HEK lysates shown in Figure 3 and suppl figure 7. In the main figure 3D authors use cells overexpressing USP18, which represents a highly unphysiological situation. It is clear that USP18 as an IFN stimulated gene is not or only marginally expressed, but overexpression clearly shifts the equilibrium. This experiment should be repeated with HEK cells or other cells stimulated with IFN to induce physiological levels of USP18. Such an approach would be a prerequisite to establish its use as a tool to monitor USP18 activity in a cellular environment.

2. Authors suggest that the developed probe represents an advanced tool for inhibitor screens. Again the Mass spec experiments were performed with cells overexpressing USP18, questioning the conclusion that "USP18 was the DUB most efficiently blocked by WP1130". Why were the cells not IFN stimulated. Alternatively, to judge specificity, not only USP18 but also other cross reactive DUBs should be overexpressed as a control.

3. The novel ABP is suggested to represent a valuable tool for AB profiling of USP18 activity in different pathologies. The authors used established lung carcinoma cell lines expressing USP18 after IFN stimulation. Results show that the probe reacts with USP18 related to its expression levels. This result is expected and does not provide novel information. Why is this not shown in relation to the wt ISG15 probe. Real tumor samples would be more relevant rather than using cell lines to support the notion that the probe is useful for monitoring USP18 activity in pathophysiological conditions.

Reviewer #2

(Remarks to the Author)

Summary

In this manuscript, Davis and colleagues detail the design and total chemical synthesis of an activity-based probe (ABPP) targeting the ISG15 deconjugating enzyme: USP18. Through careful structure-based engineering and positional scanning of a hybrid combinatorial substrate library, they devised an ABPP with high specificity for human USP18. The probe included the C-terminal part of mouse ISG15 with a substituted unnatural amino acid Agb at Arg153. They also added N-terminal acetylation/biotinylation or Cy5-conjugation depending on the application and modified the C-terminal end with a propargylamide electrophilic group. Extensive validation assays using in-gel fluorescence scanning and contemporary mass spectrometry demonstrated the ABPP's enhanced selectivity for USP18 over other DUBs, outperforming a standard wild-type ISG15-ABPP format. Additionally, the authors demonstrated two practical applications of the ABPP. One application centered on screening novel inhibitors targeting USP18, resulting in the validation of WP1130 as a USP18 inhibitor. Secondly, the authors employed the ABPP to profile USP18 activity across diverse lung carcinoma-derived cell lines which shed light on the potential for assessing the role of USP18 activity in various disease phenotypes associated with cancer.

Additionally, they authors also confirmed that ISG15 can be cleaved by other deconjugating proteases such as USP5, USP14 and USP16 as previously reported.

Relevance

USP18 is a very important regulator of IFN-I signaling but also the main and most characterized ISG15 deconjugating enzymes similar to USP2 for ubiquitin. Several reports have shed light on the therapeutic potential of USP18 mainly in the context of viral infection but also in cancer. Despite a potential clinical application, there is no validated small molecule inhibitor to block USP18 protease activity. To identify novel lead inhibitor for USP18, several ABPPs have been designed using rhodamine-USG15 assay, for instance [PMID: 32039148]. However, those assays are limited to in vitro assays. Cell lysate containing other ISG15-specific protease such as USP5, USP14 and USP16 prevent to delineate the protease activity of USP18. This is partly due to the absence of ABPP to study USP18 activity towards ISG15 in complex cell lysates. Here the authors generated the first ABPP to study USP18 activity towards ISG15 in complex cell lysates. Additionally, this assay allows to distinguish whether the inhibitors are cross-reactive for the other ISG15 specific proteases (USP5, USP14 and USP16) in addition to testing whether the inhibitors react with ubiquitin-specific DUBs (USP22, USP33 and USP42).

Major comments

Major comment 1:

The authors should comments on the choice to produce mouse ISG15 instead of human ISG15. The authors have developed mISG15CTD-based probes for selective detection of USP18. However, the rationale behind the choice of mouse ISG15 over human ISG15 for the development of a probe targeting human USP18 remains unaddressed. Likely, chemical synthesis for human isg15 has not been published (or established in the lab) while the chemical synthesis for mouse ISG15 is. Although the authors have investigated this in figure sup fig. 1, the authors should justify that mouse ISG15 is not expected to be an issue for this ABPP. However, using mouse ISG15 should remain a concern and human ISG15 should be tested as part of their validation strategy. Indeed, mouse ISG15 and human ISG15 evolved a different signaling mechanism. Human ISG15 binds to the interferon receptor through USP18 and this is essential for down regulation of interferon signaling while mouse ISG15 is obsolete for interferon signaling [PMID: 27193971; PMID: 25307056].

Major comment 2:

The specificity of ISG15-Agb lacks some evidence. This should be demonstrated in other cell line with low expression level of USP18 (e.g. A549 cells without IFN-treatment). This should be tested with biochemical and mass spectrometry-based approaches as done in fig. 3D, fig 4F-G and sup fig. 7. Indeed, the available data suggests that USP16 may have some remaining reactivity for ISG15-Agb. May be, this an artefact due to high expression of USP16 in HEKT293 cells (Fig 3D and sup fig 7B, right panel) and this would be less of an issue in other cell lines. The authors should address the remaining reactivity of ISG15-Agb to USP16. Figure 4F-G and sup fig 7B should also show the labels for USP5, USP14 and USP16.

Major comment 3:

The report is lacking a clear piece of evidence showing that inhibition of USP18 has a functional effect. This could be done at the molecular and cellular level. At the molecular level, the authors could show that WP1130 blocks the activity of USP18 in cell lysate (as evidence by increased ISG15-conjugated proteins, i.e. ISG15 smear? And no effect on the ubiquitin smear? or the activity of other DUBs: USP2, USP5, USP14 and USP16.). While at the cellular level, the authors could show that WP1130 influence antiviral defense or cancer proliferation.

Major comment 4:

Importantly, the impact of this work seems rather limited. Several layer of work are missing. The authors identified WP1130 from a small set of small molecules. The authors could increase the impact of their work by screening for a larger library of small molecules. Additionally, to increase impact, new data could show that editing the molecular structure of WP1130 can conserve activity towards USP18 while reducing activity over ubiquitin DUBs (USP22, USP33 and USP42). Finally, in vivo data would also increase the impact of this work by demonstrating that this ABPP can discover relevant inhibitor for USP18.

Minor comments

1. The introduction provides a comprehensive overview of ISGylation, highlighting the reported ISGylation of over 300 proteins in IFN-stimulated cells without a consensus sequence. While the cited papers are valuable, I recommend the authors to consider consulting the thorough review by Thery et al, which offers a comprehensive catalogue of reported substrates until 2022. Additionally, the recent work by Zhao et al offers further insights into the lysine selectivity of HERC5 [1]. Although no consensus sequence is reported, it might be beneficial to adjust the phrasing in the manuscript to "without a clear consensus sequence".

2. The authors highlight the multifaceted roles of ISGylation in maintaining genome stability, cytoskeleton dynamics, autophagy, protein translation, and hypoxia/ischemic responses. However, it is crucial to acknowledge the extensively researched function of ISGylation as a pivotal host defense pathway against both viral and bacterial infections, which seems overlooked in the current manuscript. Integrating this aspect would provide a more comprehensive understanding of ISGylation's significance in cellular physiology and immunity.

3. The authors have developed mISG15CTD-based probes for selective detection of USP18. However, the rationale behind

the choice of mouse ISG15 over human ISG15 for the development of a probe targeting human USP18 remains unaddressed.

4. This study introduces a novel activity-based probe specifically tailored for USP18, showcasing enhanced specificity compared to prior ABPPs targeting the same enzyme. While the probe's ability to screen for inhibitors within complex protein lysates represents an advancement, its significance in the broader field remains unclear. Notably, previous reports have documented the existence of USP18 ABPPs [PMID: 32039148]. While this ABPP offers advantages in complex lysate contexts, its added value in high-throughput inhibitor screening compared to using recombinant human USP18 in vitro is not readily apparent.

5. Throughout the manuscript, the authors use HEK293T lysates with or without overexpressed USP18 to assess the specificity of the ABPP. However, would it not be more relevant and fair to evaluate this in cells that express and induce endogenous levels of USP18. For example, why did the authors not use A549 cells treated with type I interferons, a condition they used in a later stage to profile USP18 activity?

6. To extend the claim that the APBB can be used to screen for specific inhibitors of USP18, I recommend including an experiment screening a small molecule library for novel USP18 inhibitors. As an alternative, the authors could screen all the inhibitors from the ISG15-rhodamine assay (sup fig. 8 right panel).

7. Furthermore, the decision to exclusively measure USP18 activity based on its truncated form raises questions. It is essential to consider that the truncated form localizes to the nucleus rather than the cytoplasm, potentially possessing distinct activity or functions from cytoplasmic USP18 (PMID: 36646704). Following this, read-out of activity relying on in-gel fluorescent detection raises concerns regarding its low-throughput nature and susceptibility to experimental variation, rendering it unsuitable for high-throughput screening across multiple cell lines (PMID: 37692766). Could the authors envision a MS-based proteomics approach?

8. It would be interesting to profile the USP18 activity in cells infected with various pathogens which could provide novel insights into the relevance of USP18 activity during infection.

9. The authors claim there is increased activity of USP18 in H2030 cells compared to the other cell lines. An experiment using the validated WP1130 inhibitor could confirm this observation and evaluate whether this has any biological significance.

10. Figure 5b. The graph shows normalized intensity of fluorescent protein band corresponding to the USP18sf_probe conjugate. However, no information is available on how the normalization was done.

11. Supplementary Figure 4. Why is there no signal of the mISG15-PA probes in the input (= left side of the gel).

12. I often see smearing of the probes on gel (compared to WT-ISG15-PA). Is this related to the chemical synthesis which is not 100% pure or are the functional groups on the probe interfering with normal running on gel?

13. Why is there always a ladder pattern for the pure ubiquitin-PA probe ; and not for the ISG15-PA probes? (e.g. Fig. 4E)

14. Sup Fig 1. The band should be quantified to appreciate to which extent USP18 is more specific toward ISG15 rather than Ubiquitin; indeed, it is hard to appreciate the bands intensity especially since the input amount of human ISG15 (FL), ubiquitin and ISG15-ctd is different (bands annotated as "ISG15-VS" and "Ub-VS" on the left side of the blots. The author could show the intensity ratio between the red star and the bands corresponding to ISG15-VS and Ubiquitin-VS. And, do the same for ISG15-ctd-VS and mouse ISG15. In this way, the increased activity of USP18 on ISG15 rather than ubiquitin will be more clear. Other quantification means would also be appreciated (unmodified USP18 versus modified-USP18?).

15. Figure 1E: Why Isoleucine was not selected in Figure 2E? It seems that it gives a strong response.

16. Here the authors wrote " In a recent study, an ISG15-based ABP was used to detect deISGylating enzymes in human cell lysates and identified USP16 as an ISG15 cross-reactive DUB in addition to the previously-characterized DUBs such as USP5 and USP14 – REF 29-32" Despite the authors mentioned "in a recent study" the authors cited several publications. They authors should clearly mention which studies, they want to highlight.

17. Figure 3A-B: the authors only tested USP5 while previously they tested USP5 (Figure 3B and sup fig.; 1B). USP2 could also be tested. Also, only mouse ISG15 was tested against human USP5. Human ISG15 could also be tested against human USP5 which is more relevant. Why did the authors omitted this test. Same comment for sup fig. 1

18. Figure 3 each blot should be quantified to be able to quantify USP18 activity over the ISG15 mutants.

19. The authors said: "We included an H90F mutant of mISG15CTD (a mutation of mISG15CTD not in the tail region) that was computationally predicted to increase binding to USP18 through hydrophobic interaction." But they do not explain the rationale or show data on this; was this derived from a published paper or was this predicted based on structural data? AlphaFold data?

20. Sup Figure 4 the blot is lacking an immunoblot showing the absence of USP18. The authors could show a similar figure as in Fig 3C with the Cy5 above and USP18 IB below. The authors should also include a positive control showing USP18 expression simply with HEK293T overexpressing FLAG-USP18 while HEK293T would show no SUP18 expression.

21. "The Ub probe demonstrated clear labeling of many proteins, while the WT mISG15CTD probe showed comparatively less but still strong labeling of several proteins as expected." The authors should mention clearly that the proteins expected are USP5, USP18 etc.. Or would the authors expect other proteins to be labeled with their probe (e.g. E1 enzymes: UBA1 or UBA7/UBE1L). The main conclusion here is that the background pattern is the same between WT and the ISG15 mutants (Arg153Agb) is the same showing that the ISG15 mutants similarly bind to human proteome.

22. Fig 3C, it would be interesting to evaluate whether the bands correspond to USP5, USP14 and USP16 or other DUBs?

23. "With no detectable basal expression of USP18, Arg153Agb and Arg153Phe(guan) mutant probes showed decreased background labeling relative to the WT, whereas the H90F mutant probe showed increased background labeling as noted by a more intense band with molecular weight near 76 kDa." The authors should clearly mentioned the figure number. (Sub figure 4?)

24. "With no detectable basal expression of USP18, Arg153Agb and Arg153Phe(guan) mutant probes showed decreased background labeling relative to the WT," The authors should quantify the blot to support their claim. They should show that the decreased bands are specific for the other DUBs (USP2, USP5, USP16). Otherwise, it is hard to speculate that the labeling reduction of this band corresponding to decrease labeling of relevant USPs (USP2, USP5, USP16).

25. Sup Fig 4 does not show bands for ISG15 but only for ubiquitin in the left side of the blot. This suggests that there is an issue with the input samples. Low exposure and high exposure blot could be used. Or this should be justify.

26. Sup Fig 4, the "H90F ISG15 mutants" is expected to bind more to USP18 as mentioned : "that was computationally predicted to increase binding to USP18 through hydrophobic interaction." However, the cell line tested is lacking USP18. So the increased band might be the evidence that the H90F ISG15 is binding to other DUBs except USP18. This is confusing since the H90F was expected to increase binding to USP18 and not other DUBs. The authors should comment on this.

27. " In the USP18WT overexpressing HEK293T cell lysates, two additional protein bands appeared, corresponding to covalent adducts of two USP18 isoforms (red arrows in Fig. 3C)". this very relevant and very nice to compare with the USP18 mutants (C64A)

28. "which was elaborated by western blot analysis." Statement is not clear and should be rephrase.

29. Sup Fig 5 , the legend is not clear while the data is better explained in the text.

30. "From preincubation of USP18WT overexpressing HEK293T cell lysates with either the WT or Arg153Agb mutant of Ac-mISG15CTD-PA prior to labeling with Cy5- mISG15CTD[WT]-PA, we observed that the R153Agb mutant selectively blocked the USP18 labeling while the WT probe competed for other proteins in addition to USP18 (Supplementary Fig. 5), again confirming enhanced selectivity of Arg153Agb for USP18 compared to the WT." The authors should re-phrase the data is not clear. What are the bands to compare? It seems that 10 μ M of WT generated less bands at 76 and 102 kDa. Could it be that the labeling is swapped?

31. Sup Fig 6A-B is clear and convincing.

32. Sup Fig 6C suggest that the mutant ISG15-Agb binds less to USP18 compared to WT USP18 (top right panel UB: USP18)? However, this data clearly demonstrates that no more USP14 bind to ISG15-Agb. This is a strong compelling case for the specificity of ISG15-Agb to USP18. However what about USP5 (tested in sup Fig 6A-B or the others DUBs?). Although this question is address in Sup Fig 7B. it would be nice to confirm this by western-blot. Additionally, if the authors test USP14 by western-blot and USP5 by in vitro assay in the sup fig 1, they should be consistent and always test SUP5 and USP14 across the whole manuscript.

33. Fig 3D and sup Fig 3B are strong evidence for the enhance specificity of ISG15-Agb. This is very compelling. However as the authors mentioned, the expression level of the DUBs play a role in the binding to the probe. Did the authors consider analyzing the proteome of HEKT cells +/- USP18 overexpression to check for the expression level of the DUBs and USP18. Alternatively, the authors could perform a competition assay in vitro to check whether at the same concentration ISG15-Agb binds more to USP18. They could mix all the DUBs: USP5, SUP14, USP16 and USP18. at equimolar concentration and check which DUBs bind most to ISG15 WT versus Agb.

34. " As expected, USP18 was the protein most highly enriched by the R153Agb probe followed by USP16 (Fig. 3D)" while USP5 and USP14 do not bind to ISG15Agb, USP16 seems to retain binding Fig 3D. Is this an artefact due to high expression of USP16 in HEKT293 cells? May be this is less of an issue in other cell lines. The authors should address the remaining reactivity of ISG15-Agb to USP16.

35. "Overall, the ABPP data recapitulate the selectivity profile of our USP18 ABP and establish its utility in quantitative chemoproteomics as a chemical tool to selectively report USP18 activity in various cell lines." The authors demonstrated that in cell lysate overexpressing USP18 the ISG15-Agb is more specific for USP18. However, this is dependent on the expression level of USP18 and the other DUBs. The authors should repeat this at a physiologic level of USP18 such as in A549 cells or other cell lines. It would be nice to check the expression level of USP18 along with USP2, USP5, USP14 and USP16. Even if challenging, the expression level should be analyzed by MS but also by Western-blot (although comparing the expression level by western-blot is far less accurate since the detection rely on different primary antibodies)

36. Figure 3D, other proteins are also enriched as strong as overexpressed USP18. The authors should comment on those. What are they? Also DUBs or other proteins?
37. Sup Fig 8 shows that PR-619 also inhibits USP18 activity similar to WP1066, why not studying this inhibitor as well?
38. Fig 4D, nice data showing that ISG15-Agb is more specific for USP18. And, that WP1130 inhibit USP18 only but not the other DUBs (as can be seen with WT ISG15). However, the blot showing USP18 expression (loading/expression control) is lacking. As mentioned above, it would also be nice to show that the bands disappearing are actually USP5, USP14 or USP16. This could be readily performed with a western-blot analysis.
39. Fig 4E, the data nicely shows the dose dependence effect (difference between 100 vs 50) but it also shows that WP1130 do not inhibit ubiquitin-specific DUBs as most bands remain with the same intensity. This is also showed with the blot against USP14 showing that WP1130 does not inhibit USP14 binding to ubiquitin. However, the authors mentioned above that WP1130 was found to inhibit USP14 activity towards ubiquitin: "However, we found that several reported DUB inhibitors, including WP1130 (a known inhibitor of USP5, USP9X, USP14, USP24, and UCH37, Fig. 4B" However, the data in Fig 4E shows that the reverse where actually USP14 is not inhibited by WP1130. The authors should comment on this.
40. Fig 4F-G, this is a very elegant approach to investigate cross-reactive inhibitors. Additionally, the authors could show all the identified DUBs as a heatmap, this will help to appreciate that USP18 is the most inhibited while few other DUBs are also inhibited by WP1130.
41. Fig 4F-G, dots are colored according to DUBs family. This should be mentioned in the legend or on the graph. The dots corresponding to the DUBs could be bigger to help distinguish the colors.
42. Sup Fig 9B the scale is indicated but it is not clear what are the values about. Is this fold change, intensity or z-score?
43. "Although not specific, we thus report WP1130 as a promising, first example of small-molecule USP18 inhibitors w" is that true? No this is not true. The first small molecule inhibiting USP18 protease activity over ISG15 was reported already [PMID: 25639862]. The authors should mention the paper. The authors could claim that WP1130 is the first strong inhibitor of USP18 but this should be supported with data showing the IC50.
44. "Nonetheless, our study demonstrates that utilizing this USP18 ABP in conjunction with a Ub-based DUB probe in a chemoproteomic screening platform holds promise as a tractable workflow for global, competitive profiling of DUB inhibitors to perform simultaneous hit identification and selectivity assessment" I fully agree with this statement. Additionally this can be done in a cell lysate compared to other state of the art approach. One perspective would be to perform this screen in cellulo. However crossing the cell membrane might be challenging although several methods have been envisioned and tested.
45. "Although we used the USP18 overexpressing system as a model, this screening approach can be applied to native or disease models of interest." Indeed, it is necessary to use more relevant model without overexpression. Alternatively, the expression level of USP18 in HEK293T cells overexpressing could be lowered by transfecting less plasmids. It would be worthwhile to show whether HEK293T cells overexpressing USP18 express USP18 close to physiologic level.
46. "A549, H358, H2030, and H1650 lung cancer lines were selected and first evaluated for their USP18 expression" the authors should mention which cell line match to the cell line name in sup fig 10.
47. "A549, H358, H2030, and H1650 lung cancer lines were selected and first evaluated for their USP18 expression" abbreviation of "interferon" to "IFN" is missing. The authors could mention the concentration and time of interferon treatment this is relevant to evaluate whether the interferon treatment was performed with experimental conditions similar to physiologic level. Ideally this is mentioned in U/mL.
48. Fig 5B, the WB against USP18 is essential to evaluate the effect of ISG15-Agb. Again, here the authors should show quantification of the western-blot. The quantification of the Cy5 is shown on the right side but it would be better and more specific to show the same quantification using the western-blot data against USP18. Additionally, USP5, USP14 and USP16 could also be tested by western-blot. When relevant the experiments should be performed with replicates to evaluate the distribution of the data.
49. "The fluorescent intensity of the USP18sf (truncated isoform)-probe conjugate band measured by densitometry indicated a slightly higher activity of USP18 in H2030 cells" the authors should indicate "USP18sf" in the figure.
50. What is the difference between Fig 5 B and sup fig 11? Is this an independent experiment repeat?
51. "...in which USP18 is highly activated and provide insights into posttranslational regulation of USP18" the authors should explain and may be show case how information about USP18 PTM can be retrieved with their approach. They should show more data.
52. Figure 5 could also show whether the probe ISG15-Agb reacts with USP5, USP14 and USP16 (+/-IFN). This should be tested with a "high exposure" western-blot against USP18 like in Fig 5A middle panel. It would also be nice to test the WP1130 in those cell lines as well.

Typos

In the text below: “WP”, “Quench” and “Final” are the words with a typo.

“Quantitative MS-based ABPP HEK 293T cells were lysed (50 mM Tris pH 8.0, 150 mM NaCl, 5 mM MgCl₂, 0.5 mM EDTA, 0.5% NP-40, 10% glycerol, 1 mM TCEP, protease inhibitor (cOmplete™ Mini Protease Inhibitor Cocktail, Roche)), clarified by centrifugation, and then diluted to a concentration of 2 mg/mL. 500 µL aliquots were incubated with 50 or 100 µM of the WP or DMSO for 2 h at room temperature. Subsequently, 1 µM of Ub or EY-2-137(Agb) probe were treated for 3 h at room temperature. To Quench the reaction, 500 µL of 0.2% SDS, 0.5% NP-40 in PBS were added to cell lysate (Final 0.1% SDS, 0.25% NP-40).”

In the abstract, the following text: “Combining with a ubiquitin-based DUB ABP, the selective USP18 ABP is employed in a chemoproteomic screening platform to identify and assess inhibitors of DUBs including USP18”; The words “Combining” should be “combined”.

Reviewer #3

(Remarks to the Author)

Reviewer #4

(Remarks to the Author)

New probes that can detect the different DUB activities are still needed. This is especially true for the studying the ISGylation activity of USP18. I really have nothing to add to this paper other than they should include a few more references for the screening as this peptide MS technique has been done by others, especially for proteasome related probes. The data is well presented and proper controls have been included.

Version 1:

Reviewer comments:

Reviewer #1

(Remarks to the Author)

The authors have thoroughly addressed my concerns and performed additional experiments which clearly increased the overall quality of this manuscript. Although the experiment I asked for (Mass spec in IFN cells not overexpressing USP18) was not performed, within the context of the new data, given arguments about the general focus of this work (application spectrum) and altered discussion this work is now conclusive and clear.

I also agree that using the probe to evaluate USP18 activity in primary tumor material might be better addressed systematically in a follow up study. By addressing the action of the ABP towards crossreactive USPs profound additional information is given.

Reviewer #2

(Remarks to the Author)

I appreciate the authors' efforts to address the concerns raised during the initial review. The additional experiments conducted provide significant added value to the manuscript. I am pleased to see that two of my major comments have been addressed. However, before the manuscript can be considered for publication, two of my major comments remain to be addressed along with a few minor comments (see below). Overall, I am pleased with the improvements made to the manuscript and look forward to reviewing the revised version.

Resolved major comments:

- The use of mouse ISG15 instead of human ISG15 for the probe is motivated by the dual specificity of mouse ISG15 for human and mouse USP18 along with technical challenges of human ISG15).
- The design of USP18 inhibitors with high specificity and affinity is beyond the scope of this manuscript and will be addressed in future work.

Remaining major comments to be addressed:

- Determining the reactivity of the mISG15ctd[R153Agb] probe with USP16 (rather than USP18). The authors demonstrated that the ISG15 R153Agb mutant binds less to USP16 compared to WT ISG15, but they fail to demonstrate that USP18 binds more to the ISG15 R153Agb mutant compared to WT ISG15 (see below). The authors could determine affinity of USP18 and USP16 for WT ISG15 and ISG15 R153Agb mutant.
- Investigating the functional effects of USP18 inhibition remains to be addressed. The claim that WP1130 inhibits USP18 in

cellulo is not convincing (see below).

Major comments 1:

- Major Comment 1: Determining the reactivity of mISG15ctd[R153Agb] probe with USP16

- I thank the authors for the new data provided to investigate the cross-reactivity of the ISG15 R153Agb mutant toward USP16. I believe this is of clear added value and demonstrates that the ISG15 R153Agb mutant has reduced binding to USP16 compared to WT ISG15. However, this data (Fig 7) fails to demonstrate the superior specificity of the ISG15 R153Agb mutant to USP18 compared to USP16. Instead, the data in Figure 5C suggests that the ISG15 R153Agb mutant has dual specificity for USP18 and USP16, likely to the same extent.
- Data showing the contrary would be crucial to support the main claim of the paper. One piece of evidence could be to determine the binding affinity of the ISG15 R153Agb mutant (and WT ISG15) to USP18 and USP16. Alternatively, the authors could design a chemoproteomics screen to resolve the issue of this dual specificity toward USP18 and USP16. One could propose to deplete the cell lysate of USP16 by immunoprecipitation or genetic engineering (KO cells?). Using such an approach, the ISG15 R153Agb mutant probe would be fully specific to USP18.
- Similar to the above, Figure 5C clearly shows that ISG15 R153Agb binds less to USP5, while the binding to USP18 is retained. This demonstrates the increased specificity of ISG15 R153Agb toward USP18 compared to USP5, as desired. However, unlike USP5, USP16 retains binding to ISG15 R153Agb mutant similar to USP18. This data raises questions about the exclusive selectivity of ISG15 R153Agb mutant toward USP18, as proposed by the authors. Thus, the data in Fig 5C suggests that ISG15 R153Agb mutant probe has dual specificity toward USP18 and USP16.
- Noteworthy, the data in Figure 5C contradicts the data in Sup Fig 7. Figure 5C shows that WT ISG15 and ISG15 R153Agb mutant bind to USP16 to the same extent in A549 cells, while ISG15 R153Agb mutant binds less to USP16 compared to WT ISG15 in Sup Fig 7. Can the authors comment on this discrepancy? How do the authors explain that USP16 binds to ISG15 R153Agb mutant and WT ISG15 to the same extent? One could suppose that the results can be explained by the higher expression level of USP16 in A549 cells compared to HEK293T. Consequently, the pulldown of ISG15 R153Agb mutant enriched the same amount of USP16 as the pulldown of WT ISG15 despite USP16 has higher affinity for WT ISG15 and less affinity for ISG15 R153Agb mutant (fig 7). This is a well-known principle where the enrichment of prey protein (here USP16 or USP18) by pulldown is driven by affinity to the bait (here ISG15 R153Agb) but also by the concentration of the prey. Here again, it would be worthwhile to determine the abundance of USP16 and USP18 by shotgun proteomics and IBAQ intensity to explain the different results in Fig 5C and sup fig 7.

- Major Comment 2: Investigating functional effects of USP18 inhibition

- The inhibition of USP18 by WP1130 is clearly not supported by the data. The increase in ISGylated proteins in the insoluble fraction is more likely due to cellular stress or toxicity (or another biological phenomenon: autophagy induction through p62 or HDAC6?), rather than USP18 inhibition.
- In this experiment (Fig 6), the authors chose to perform the drug treatment with WP1130 on living cells, while in the previous experiments (Fig 4 and 5), they performed the treatment in a cleared cell lysate. To demonstrate USP18 inhibition by WP1130, the authors could also perform the drug treatment in cell lysate and monitor deISGylation activity by western blot to detect ISGylated proteins. This has been done elsewhere using catalytically inactive USP18 versus wild-type (PMID: 28165509, Figure 2C). Showing this data will demonstrate functional effect at the molecular level but not at cellular level indeed. Functional effects at cellular level could be addressed in future work.

Minor comments:

- For the activity-based probe profiling (ABPP) in lysate, the reaction volume should be mentioned. The method section only contains the protein amount and probe concentration. This information is important if one wishes to repeat the assay.
- Regarding the use of USP18 ABP under physiological conditions: "One of the advantages of using the probe more specific for USP18 in the ABPP approach when combined with Ub-based ABP is that we avoid significant competition between probes for DUB profiling," said by the authors, but this strategy is unclear. One alternative approach would be to deplete the lysate of "ubiquitin-binding DUBs" with the ubiquitin probe as a first step. Then, in a second step, this "ubiquitin-cleared" lysate could be used for the mISG15ctd[R153Agb] probe. The authors should comment on this strategy or mention this alternative in the text.
- The authors said, "Although R153Agb mutant probes are more specific for USP18 over other ISG15-crossreactive DUBs compared to existing WT ISG15-based probes." What are the data supporting this claim? To support this statement, the authors should report binding affinity (K_d , K_{on} , or K_{off}). Alternatively, the authors can rephrase this statement. While the authors show that the R153Agb mutant probe is less reactive to USP5 and other DUBs compared to WT ISG15, they did not show that the R153Agb mutant is more specific for USP18 compared to other DUBs. There is a subtle difference in the phrasing. This was already mentioned above as major comments but please re-phrase the sentence.
- The authors said, "Our chemoproteomics data indicated that they react with USP16 to some extent, especially when USP18 is expressed at a relatively low level." This statement should be supported with some data. Is this referring to the expression level of USP18 in A549 cells (Fig 5C)? The expression level of USP18 seems unknown from the manuscript. Could the authors show the expression level of the DUBs in the lysate of A549 +/- interferon? (and HEK293T cells?) The authors could perform shotgun proteomics on those samples and show protein rank as before. Expression level can be

evaluated using IBAQ intensity values. Although not perfect, IBAQ intensity allows an estimation of protein expression and comparison between proteins. Alternatively, the authors could make use of proteomics or transcriptomics data already published elsewhere.

- Figures 7B and 7C should be quantified and represented with a bar plot.
- The sentence here is unclear please rephrase: "Comparing other proteins detected by R153Agb relative to the WT, USP16 was found to be ranked high while enrichment of USP5, USP14, USP30, and USP35 was significantly diminished, indicating the R153Agb probe reacts with USP16 when its target protein USP18 is expressed at a relatively low level".
- In Fig 5D, the legend should be updated to clarify that the heatmap was made based on results from mixed probes (ISG15 probe and Ub probe)—this is clearly stated in the text, but not in the legend.
- "One of the advantages of using the probe more specific for USP18 in the ABPP approach when combined with Ub-based ABP is that we avoid significant competition between probes for DUB profiling." This statement is unclear please rephrase. The pulldown combining ubiquitin and ISG15 as prey is an interesting approach but the rationale should be explained further in the text. (This point was mentioned earlier)

Reviewer #3

(Remarks to the Author)

Reviewer #4

(Remarks to the Author)

All questions from the previous review have been addressed. Congrats on the paper.

Version 2:

Reviewer comments:

Reviewer #2

(Remarks to the Author)

I thank the authors for performing additional experiments. Almost all my previous comments have been addressed. I have one final minor comment (see below). Upon completion, I recommend publication of this manuscript.

In their answer, the authors indicate that they do not claim that the ISG15Agb has more specificity for USP18 over USP16. But, this seems to be essential for designing an ISG15 probe specific for USP18 over other DUBs (e.g. USP16) as mentioned in the title. How do the authors intend to study exclusively USP18 then? To do so, one can increase the affinity of the ISG15 probe for USP18. Alternatively, one can deplete the ISG15-cross reactive DUBs by genetic knock-out or immunoprecipitation. The authors proposed to covalently block all ubiquitin DUBs prior using the ISG15 probe (sup fig 15). This another elegant strategy to reach the same goal. This approach is convincing however the sup fig 15 is lacking the immunoblotting for the ISG15 cross-reactive DUBs, especially USP16. The indication of the molecular weight is not sufficient.

The data in fig 6A is convincing. I thank the authors for performing this additional experiment. This demonstrates the inhibitory activity of WP1130 on ISGylated proteins that were produced at endogenous level.

Reviewer #3

(Remarks to the Author)

Response to reviewers' comments

Thank you for the expert reviews of our manuscript entitled "Chemical tools to define and manipulate interferon-inducible Ubl protease USP18 (NCOMMS-24-23167)". We are grateful for the overall positive response and substantive comments. These comments were well-taken and have prompted us to perform additional experiments and revisions to the text. Enclosed is a summary of the key changes, a point-by-point response, and the revised manuscript (including the track changes version) and the supporting information. Overall, these changes address the reviewers' concerns and have greatly strengthened the manuscript.

The most significant points raised by the reviewers and our response are summarized as follows:

1. Validating the utility of USP18 ABPs in physiologically more relevant conditions

Reviewer 1 raised an important question regarding the utility of our USP18 activity-based probes (ABPs) not only in the overexpression system but also in cells stimulated with IFN to measure USP18 activity in physiologic environments. We agree with the reviewer's point and that is the reason why we performed the activity-based profiling of USP18 in lung cancer cell lines after induction of USP18 expression with IFN stimulation using Cy5-mISG15_{CTD}-PA probes and in-gel fluorescence scanning (Fig. 5B and Supplementary Fig. 12). As suggested by the reviewer, we now have performed proteomic analysis of streptavidin pulldowns with lysates prepared from A549 cells after IFN stimulation to further validate the efficiency and selectivity of our probes. The mISG15_{CTD}[R153Agb] biotin probe was found to efficiently enrich USP18. Combined with the Ub probe, we were able to detect USP18 along with other 44 DUBs. These new data are now included in Fig. 5C, Supplementary Fig. 14, and Supplementary Table 4. Additionally, we carried out competitive activity-based protein profiling (ABPP) to assess the inhibition of USP18 by WP1130 in A549 cell lysates (Fig. 5D, Supplementary Fig. 15, and Supplementary Table 5). At two different concentrations we tested (50 and 100 μ M), WP1130 was shown to inhibit USP18 and several other DUBs, most notably USP9X, in A549 cells. One of the advantages of using the probe more specific for USP18 in the ABPP approach when combined with Ub-based ABP is that we avoid significant competition between probes for DUB profiling. Taken together, these new data, we believe, demonstrate the utility of USP18 probes, with improved selectivity, in profiling the activity of USP18 in physiologically relevant environments.

2. Determining reactivity of mISG15_{CTD}[R153Agb] probes with USP16

Reviewer 2 asked for further evidence that mISG15_{CTD}[R153Agb] probes are specific to USP18. Although R153Agb mutant probes are more specific for USP18 over other ISG15-crossreactive DUBs compared to existing WT ISG15-based probes, our chemoproteomics data indicated that they react with USP16 to some extent, especially when USP18 is expressed at a relatively low level. To determine their reactivity to USP16, we have performed gel-based protein profiling experiments using both recombinant USP16 and USP16-FLAG overexpressing cell lysates. We also prepared a fluorogenic substrate by incorporating rhodamine 110 at the C-terminal end of mISG15_{CTD}[R153Agb] that exhibits red-shifted fluorescence upon cleavage by enzymes to measure its interaction with USP16 in an orthogonal assay. All these experiments indicated that the R153Agb mutant of mISG15_{CTD} binds and interacts with USP16 but at a comparatively lower level than the WT. Therefore, the ability and sensitivity of these probes to detect USP16 or USP18 may depend on the relative abundance and activity between these two proteins in the conditions used. The new data now have been included in Fig. 7 and Supplementary Fig. 17.

3. Investigating functional effects of USP18 inhibition

Reviewer 2 asked about the evidence for functional inhibition of USP18 by WP1130. After confirming that WP1130 inhibits USP18 activity in A549 cell lysates, we examined the functional effects of WP1130 in A549 cells. WP1130 was found to be cytotoxic and antiproliferative as reported presumably through inhibition of several DUBs. To analyze the effect of USP18 inhibition by WP1130 on protein ISGylation, A549 cells were stimulated with IFN- β and treated with increasing concentrations of WP1130. The cell lysates were then analyzed by immunoblotting. We observed the accumulation of ISGylated insoluble aggregates of protein in WP1130-treated cells, suggesting the inhibition of USP18's deISGylating activity. It has been reported that ISG15 conjugation marks target substrates for interaction with HDAC6 and SQSTM1/p62 toward autophagic clearance. Our study with WP1130 suggests that inhibition of deISGylase activity of USP18

enhances cellular levels of ISGylated proteins that can trigger aggregate formation (aggresomes). The new data are included in Fig. 6 and Supplementary Fig. 16.

Below is a point-by-point response.

Comment	Response
Reviewer 1 3. The novel ABP is suggested to represent a valuable tool for AB profiling of USP18 activity in different pathologies. The authors used established lung carcinoma cell lines expressing USP18 after IFN stimulation. Results show that the probe reacts with USP18 related to its expression levels. This result is expected and does not provide novel information. Why is this not shown in relation to the wtISG15 probe? Real tumor samples would be more relevant rather than using cell lines to support the notion that the probe is useful for monitoring USP18 activity in pathophysiological conditions.	Thanks for pointing this out. The protease activity is not always correlated with its expression as it needs to be tightly regulated because of its irreversible nature. Our goal is to offer useful reagents and approaches enabling measurement of USP18 activity with improved selectivity in addition to existing methods measuring expression, which can provide insight into contexts where USP18 activity is dysregulated. We agree with the reviewer that using real tumor samples is more relevant. Indeed, our ongoing work includes USP18 activity profiling studies with tumor tissue samples using probes developed here. This current study, however, is focused on the design and validation of USP18 ABPs. To make this point clear and provide outlooks, we now have added sentences in Discussion that reads as follows: “Although we observed that the reaction of ISG15-based ABPs with USP18 is dependent on its expression level at least in cells tested in this study, since these tools measure the catalytic activity of USP18, the chemical approach presented here can provide insight into contexts where USP18 activity is dysregulated independent of its expression. Moving forward, this strategy should allow us to profile USP18 activity in various cellular and disease states, and eventually elucidate the contribution of its deISGylating activity to the disease phenotype. In particular, our future work will be focused on activity profiling of USP18 in tumor tissue samples. Combined with the Ub-based DUB activity profiling data, the ISG15-based activity profiling of deISGylases and USP18 will help comprehensively investigate and define the activity-related landscape of Ubl proteases and unravel important Ub and Ubl-dependent biological processes in cancer.” And the result from same experiments using the WT ISG15 probe was already included in Supplementary Fig. 12.
Reviewers 2 & 3 1. The authors should comment on the choice to produce mouse ISG15 instead of human ISG15. The authors have developed mISG15_{CTD}-based probes for selective detection of USP18. However, the rationale behind the choice of mouse ISG15 over human ISG15 for the development of a probe targeting human USP18 remains unaddressed. Likely, chemical synthesis for human ISG15 has not been published (or established in the lab) while the chemical synthesis for	We should have made this clearer in the original submission. The reason why we decided to use mouse ISG15 sequence for USP18 ABP development is because mISG15_{CTD} has sufficient binding to both human and mouse USP18 based on our labeling experiments (Supplementary Fig. 1A) and cross-species compatibility has also been reported for ISG15 processing enzymes. Despite some synthetic challenges of human ISG15 regarding solubility and aggregation issue, we were able to synthesize human ISG15_{CTD} probe (Ac-hISG15_{CTD}-VS) and the synthetic protocol is now included in the manuscript.

mouse ISG15 is. Although the authors have investigated this in figure sup fig. 1, the authors should justify that mouse ISG15 is not expected to be an issue for this ABPP. However, using mouse ISG15 should remain a concern and human ISG15 should be tested as part of their validation strategy. Indeed, mouse ISG15 and human ISG15 evolved a different signaling mechanism. Human ISG15 binds to the interferon receptor through USP18, and this is essential for downregulation of interferon signaling while mouse ISG15 is obsolete for interferon signaling [PMID: 27193971; PMID: 25307056].	The utility of these ABPs, however, is not necessarily to study endogenous ISG15 signaling pathway where sequence variability between species would have significant impact, especially in the context of interferon signaling, immune function, and antiviral mechanism as the reviewer pointed out. As these chemical probes are used to engage USP18 in a catalytic activity-dependent manner for profiling of protease activity, the role of ISG15 sequence for probe design is focused on its molecular interaction with USP18 as a recognition element. We agree with the reviewer that it is important to note the limitation of using mISG15_{CTD}, thus added following sentences in Discussion. “Additionally, in this study, we used a C-terminal domain of the mouse ISG15 sequence as a recognition element for designing chemical probes to profile the protease activity of USP18 because it has shown sufficient binding to both human and mouse USP18, and cross-species compatibility has been reported for ISG15 processing enzymes.⁶⁸ Our ongoing work includes the synthesis and characterization of probes using human ISG15 sequence, both a truncated C-terminal and a full length, to determine whether the same mutation incorporation would work with human sequence and understand cross-species interactions of ISG15 with DUBs as sequence variability between species may have a significant impact on molecular interactions. We have already successfully synthesized hISG15_{CTD} using solid-phase peptide chemistry and expect this established protocol will be amenable to incorporating unnatural amino acids.”
4. Importantly, the impact of this work seems rather limited. Several layers of work are missing. The authors identified WP1130 from a small set of small molecules. The authors could increase the impact of their work by screening for a larger library of small molecules. Additionally, to increase impact, new data could show that editing the molecular structure of WP1130 can conserve activity towards USP18 while reducing activity over ubiquitin DUBs (USP22, USP33 and USP42). Finally, in vivo data would also increase the impact of this work by demonstrating that this ABPP can discover relevant inhibitors for USP18.	We agree that development of more potent and selective USP18 inhibitors is of importance, and we are currently working on high-throughput screening and medicinal chemistry campaign which have already produced promising results. However, delivery of in vivo effective inhibitors is not an easy endeavor that will likely take years, and we believe is beyond scope of this current study as we focus here on an activity-based chemical proteomics strategy using newly developed tools that can accelerate inhibitor development. We now have added discussion about the limitation of WP1130 and need for the development of potent and selective small molecule inhibitors of USP18 that reads as follows: “In our initial effort using a small set of compounds, we discovered that a small molecule pan-DUB inhibitor WP1130 inhibits USP18 and enhances the accumulation of ISGylated aggregates. WP1130 displayed cytotoxic and antiproliferative properties in multiple cancer cell lines probably through inhibition of several DUBs. Mechanistic, molecular contribution of USP18 inhibition to these cellular effects, however, remains unclear. WP1130 has also been reported to exhibit potent antiviral effects against norovirus and several RNA viruses.^{69,70} As ISGylation pathway plays an important role in antiviral immunity as a defense mechanism, it would be of great interest to investigate USP18 inhibitors in viral infections. All these outstanding

	questions require more potent and selective ligands. With fluorogenic substrates and ABPs more specific for USP18 in our hands, we have launched a high-throughput biochemical and chemoproteomic screening campaign to identify small molecule USP18 inhibitors and anticipate expanding a chemical toolbox for studying (de)ISGylation pathways.”
Reviewer 4 New probes that can detect the different DUB activities are still needed. This is especially true for the studying the ISGylation activity of USP18. I really have nothing to add to this paper other than they should include a few more references for the screening as this peptide MS technique has been done by others, especially for proteasome related probes. The data is well presented and proper controls have been included.	Thanks for pointing this out. We now have included following references for chemoproteomics screening. 33. Li, N. et al. Relative quantification of proteasome activity by activity-based protein profiling and LC-MS/MS. Nature Protocols 8, 1155-1168 (2013). 34. Federspiel, J.D. et al. Specificity of protein covalent modification by the electrophilic proteasome inhibitor carfilzomib in human cells. Molecular & Cellular Proteomics 15, 3233-3242 (2016). 35. Mirabella, A.C. et al. Specific cell-permeable inhibitor of proteasome trypsin-like sites selectively sensitizes myeloma cells to bortezomib and carfilzomib. Chem Biol 18, 608-18 (2011).

Minor comments:

1. The introduction provides a comprehensive overview of ISGylation, highlighting the reported ISGylation of over 300 proteins in IFN-stimulated cells without a consensus sequence. While the cited papers are valuable, I recommend the authors to consider consulting the thorough review by They et al, which offers a comprehensive catalog of reported substrates until 2022. Additionally, the recent work by Zhao et al offers further insights into the lysine selectivity of HERC5. Although no consensus sequence is reported, it might be beneficial to adjust the phrasing in the manuscript to "without a clear consensus sequence".
- The suggested papers are now cited, and the phrase has been modified.
2. The authors highlight the multifaceted roles of ISGylation in maintaining genome stability, cytoskeleton dynamics, autophagy, protein translation, and hypoxia/ischemic responses. However, it is crucial to acknowledge the extensively researched function of ISGylation as a pivotal host defense pathway against both viral and bacterial infections, which seems overlooked in the current manuscript. Integrating this aspect would provide a more comprehensive understanding of ISGylation's significance in cellular physiology and immunity.
- Thanks for pointing this out. We have now added a sentence in the Introduction that reads as follows: The ISG15 system has been largely implicated in antiviral immunity as an important cellular defense mechanism to restrict infections since elevated ISGylation of viral or host proteins has been reported to accompany increased viral resistance.¹¹⁻¹³
3. The authors have developed mISG15CTD-based probes for selective detection of USP18. However, the rationale behind the choice of mouse ISG15 over human ISG15 for the development of a probe targeting human USP18 remains unaddressed.
- This has been addressed. Please see the response to Reviewer 2's comment 1 above.
4. This study introduces a novel activity-based probe specifically tailored for USP18, showcasing enhanced specificity compared to prior ABPPs targeting the same enzyme. While the probe's ability to screen for inhibitors within complex protein lysates represents an advancement, its significance in the broader field remains unclear. Notably, previous reports have documented the existence of USP18 ABPPs [PMID: 32039148]. While this ABPP offers advantages in complex lysate contexts, its added value in high-throughput inhibitor screening compared to using recombinant human USP18 in vitro is not readily apparent.

- We believe our new competitive ABPP data in the lung cancer cell line (Fig. 5D) demonstrate the utility of our USP18 ABPs in an adaptable, streamlined platform for accelerating DUB inhibitor discovery including USP18 in physiologically relevant cell contexts. One of the advantages of using the probe more specific for USP18 in this approach is that we can avoid significant competition between probes when combined with Ub-based ABP for DUB profiling. We are currently expanding our screening efforts as well.

5. Throughout the manuscript, the authors use HEK293T lysates with or without overexpressed USP18 to assess the specificity of the ABPP. However, would it not be more relevant and fair to evaluate this in cells that express and induce endogenous levels of USP18. For example, why did the authors not use A549 cells treated with type I interferons, a condition they used in a later stage to profile USP18 activity?

- This has been addressed. Please see the response above.

6. To extend the claim that the APBB can be used to screen for specific inhibitors of USP18, I recommend including an experiment screening a small molecule library for novel USP18 inhibitors. As an alternative, the authors could screen all the inhibitors from the ISG15-rhodamine assay (sup fig. 8 right panel).

- Please see the response to Reviewer 2's comment 4 above. New data from gel-based ABPP with selected DUB inhibitors are now included in Supplementary Fig. 9.

7. Furthermore, the decision to exclusively measure USP18 activity based on its truncated form raises questions. It is essential to consider that the truncated form localizes to the nucleus rather than the cytoplasm, potentially possessing distinct activity or functions from cytoplasmic USP18 (PMID: 36646704). Following this, read-out of activity relying on in-gel fluorescent detection raises concerns regarding its low-throughput nature and susceptibility to experimental variation, rendering it unsuitable for high-throughput screening across multiple cell lines (PMID: 37692766). Could the authors envision a MS-based proteomics approach?

- We apologize for the confusion. Our intention is not to exclusively measure the activity of the truncated USP18 isoform. To make this point clear, we amended the text and figure title in Fig. 5B. In addition to Cy5 probes, we already have developed biotin probes allowing for the MS-based proteomics approach (Figs. 3D and 5D and Supplementary Figs. 7 and 14).

8. It would be interesting to profile the USP18 activity in cells infected with various pathogens which could provide novel insights into the relevance of USP18 activity during infection.

- We agree with the reviewer that the probes developed in this study can be used to profile the USP18 activity in different infection and immune models, however, it is beyond the scope of this current study. We instead have added a sentence in Discussion that reads as follows: "WP1130 has been also reported to exhibit potent antiviral effects against norovirus and several RNA viruses.^{69,70} As ISGylation pathway plays an important role in antiviral immunity as a defense mechanism, it would be of great interest to investigate USP18 inhibitors in viral infections."

9. The authors claim there is increased activity of USP18 in H2030 cells compared to the other cell lines. An experiment using the validated WP1130 inhibitor could confirm this observation and evaluate whether this has any biological significance.

- The increased activity of USP18 we noted in H2030 cells was rather mild and not substantially different from other cell lines we tested. The point was to highlight the utility of ABPs in detecting the differential activity of USP18 in proteomes. We used WP1130 to inhibit USP18 activity in A549 cells instead and confirmed that WP1130 treatment leads to rapid accumulation of ISGylated insoluble aggregates (Fig. 6B).

10. Figure 5b. The graph shows the normalized intensity of the fluorescent protein band corresponding to the USP18sf-probe conjugate. However, no information is available on how the normalization was done.

- The intensity of the fluorescent protein band corresponding to the USP18sf-probe conjugate in each condition was normalized to the fluorescent intensity of USP18sf-probe conjugate measured from USP18 overexpressing conditions. The figure legend has been revised to include this information.

11. Supplementary Figure 4. Why is there no signal of the mISG15-PA probes in the input (= left side of the gel).

- It ran to the bottom of the gel and was cut off. The protein band that appeared on the first lane with the Ub probe (commercial) has a higher molecular weight than the 9 kDa Cy5-conjugated monoubiquitin as an additional background signal from the probe. The figure now includes the probe and sample buffer signals on the bottom of the gel.

12. I often see smearing of the probes on the gel (compared to WT-ISG15-PA). Is this related to the chemical synthesis which is not 100% pure or are the functional groups on the probe interfering with normal running on gel?

- The WT probe was also chemically synthesized in the lab and all of our probes showed >98% purity determined by LC-HRMS. We believe it could be how they run on these 4-12% Bis-Tris gels.

13. Why is there always a ladder pattern for the pure ubiquitin-PA probe and not for the ISG15-PA probes?

- It is because of more intense fluorescently labeled proteins present when treated with the ubiquitin probe compared to ISG15 probes.

14. Sup Fig 1. The band should be quantified to appreciate to which extent USP18 is more specific toward ISG15 rather than Ubiquitin; Indeed, it is hard to appreciate the band intensity especially since the input amount of human ISG15 (FL), ubiquitin and ISG15-ctd is different (bands annotated as "ISG15-VS" and "Ub-VS" on the left side of the blots. The author could show the intensity ratio between the red star and the bands corresponding to ISG15-VS and Ubiquitin-VS. And, do the same for ISG15-ctd-VS and mouse ISG15. In this way, the increased activity of USP18 on ISG15 rather than ubiquitin will be clearer. Other quantification means would also be appreciated (unmodified USP18 versus modified USP18?).

- We appreciate the comment and agree that quantification of protein band intensity would be helpful in assessing the relative reactivity of these probes with USP18. However, we do not think that would add critical value to the observation that has been also previously reported.

15. Figure 1E: Why Isoleucine was not selected in Figure 2E? It seems that it gives a strong response.

- As a combinatorial library contains a mixture of substrates, we need to synthesize individual compounds to deconvolute the one responsible for generating a response. When individually synthesized and tested, Ac-LIGG-ACC was not efficiently cleaved by USP18 (Supplementary Fig. 2).

16. Here the authors wrote, "In a recent study, an ISG15-based ABP was used to detect deISGylating enzymes in human cell lysates and identified USP16 as an ISG15 cross-reactive DUB in addition to the previously-characterized DUBs such as USP5 and USP14 – REF 29-32" Despite the authors mentioned "in a recent study" the authors cited several publications. They authors should clearly mention which studies, they want to highlight.

- This has been corrected.

17. Figure 3A-B: the authors only tested USP5 while previously they tested USP5 (Figure 3B and sup fig.; 1B). USP2 could also be tested. Also, only mouse ISG15 was tested against human USP5. Human ISG15 could also be tested against human USP5 which is more relevant. Why did the authors omit this test? Same comment for sup fig. 1.

- Our synthetic probe was developed using a mouse ISG15 sequence as it has shown sufficient binding to both mouse and human USP18. Please see our response to Reviewer 2's comment 1 above. USP5 was selected for initial counter-screening because it is the most abundant and reactive DUB to ISG15. We now include screening results with USP16.

18. Figure 3: each blot should be quantified to be able to quantify USP18 activity over the ISG15 mutants.

- Fluorescent ABPs are powerful tools for quick visualization of protein activity using gel-based assays. Although the intensity of fluorescently labeled proteins can be quantified, we used a more robust, orthogonal MS-based ABPP approach to quantify USP18 activity (Fig. 3D).

19. The authors said: "We included an H90F mutant of mISG15CTD (a mutation of mISG15CTD not in the tail region) that was computationally predicted to increase binding to USP18 through hydrophobic interaction." But they do not explain the rationale or show data on this; was this derived from a published paper or was this predicted based on structural data? AlphaFold data?

- The following sentences have been added. “Additionally, we included an H90F mutant of mISG15_{CTD} (a mutation outside of the tail region) that was computationally predicted to increase binding to USP18 through hydrophobic interactions according to structure-based computational mutational analysis utilizing FoldX software⁴⁹ and RosettaDesign server⁵⁰. Through in silico alanine and positional scanning using the crystal structure of the mUSP18-ISG15 complex (PDB ID: 5CHV)⁴⁵, we identified key interface residues that optimize the interaction between ISG15 and USP18.”

20. Sup Figure 4 the blot is lacking an immunoblot showing the absence of USP18. The authors could show a similar figure as in Fig 3C with the Cy5 above and USP18 IB below. The authors should also include a positive control showing USP18 expression simply with HEKT overexpressing FLAG-USP18 while HEK293T would show no USP18 expression.

- Immunoblotting data is now included in Supplementary Fig. 4.

21. “The Ub probe demonstrated clear labeling of many proteins, while the WT mISG15_{CTD} probe showed comparatively less but still strong labeling of several proteins as expected.” The authors should mention clearly that the proteins expected are USP5, USP18 etc.. Or would the authors expect other proteins to be labeled with their probe (e.g. E1 enzymes: UBA1 or UBA7/UBE1L). The main conclusion here is that the background pattern is the same between WT and the ISG15 mutants (Arg153Agb) is the same showing that the ISG15 mutants similarly bind to human proteome.

- The sentence has been modified that read as follows: “In our labeling experiments, the Ub probe demonstrated clear labeling of many proteins, while the WT mISG15_{CTD} probe showed strong labeling of fewer DUBs, such as USP5, USP16, and USP14 based on the molecular weight, as expected.”

22. Fig 3C, it would be interesting to evaluate whether the bands correspond to USP5, USP14, and USP16 or other DUBs?

- Our MS analysis of pulldowns with biotin probes indicated that they are likely USP5, USP14, and USP16 as they were most highly enriched.

23. “With no detectable basal expression of USP18, Arg153Agb and Arg153Phe(guan) mutant probes showed decreased background labeling relative to the WT, whereas the H90F mutant probe showed increased background labeling as noted by a more intense band with a molecular weight near 76 kDa.” The authors should clearly mention the figure number. (Sub figure 4?)

- The figure number has been added.

24. “With no detectable basal expression of USP18, Arg153Agb and Arg153Phe(guan) mutant probes showed decreased background labeling relative to the WT,” The authors should quantify the blot to support their claim. They should show that the decreased bands are specific for the other DUBs (USP2, USP5, USP16). Otherwise, it is hard to speculate that the labeling reduction of this band corresponding to decrease labeling of relevant USPs (USP2, USP5, USP16).

- Although the intensity of fluorescently labeled proteins can be quantified, we used a more robust MS-based ABPP approach to identify and quantify reactive DUBs and support this claim (Fig. 3D).

25. Sup Fig 4 does not show bands for ISG15 but only for ubiquitin in the left side of the blot. This suggests that there is an issue with the input samples. Low exposure and high exposure blot could be used. Or this should be justify.

- It ran to the bottom of the gel and was cut off. The protein band that appeared on the first lane with the Ub probe (commercial) has a higher molecular weight than the 9 kDa Cy5-conjugated monoubiquitin as an additional background signal from the probe. The figure now includes the probe and sample buffer signals on the bottom of the gel.

26. Sup Fig 4, the “H90F ISG15 mutants” is expected to bind more to USP18 as mentioned: “that was computationally predicted to increase binding to USP18 through hydrophobic interaction.” However, the cell line tested is lacking USP18. So the increased band might be the evidence that the H90F ISG15 is binding to other DUBs except USP18. This is confusing since the H90F was expected to increase binding to USP18 and not other DUBs. The authors should comment on this.

- The H90F mutation was predicted to increase binding to USP18 but not necessarily selectivity to USP18 over other DUBs and our data show that it does bind to ISG15-crossreactive DUBs.

28. "which was elaborated by western blot analysis." The statement is not clear and should be rephrased.
- This has been rephrased to "which was elaborated by western blotting analysis where both unconjugated and conjugated full length and truncated USP18 were detected."

29. Sup Fig 5, the legend is not clear while the data is better explained in the text.
- The legend has been modified.

30. "From preincubation of USP18WT overexpressing HEK293T cell lysates with either the WT or Arg153Agb mutant of Ac-mISG15^{CTD}-PA prior to labeling with Cy5- mISG15^{CTD}[WT]-PA, we observed that the R153Agb mutant selectively blocked the USP18 labeling while the WT probe competed for other proteins in addition to USP18 (Supplementary Fig. 5), again confirming enhanced selectivity of Arg153Agb for USP18 compared to the WT." The authors should re-phrase the data is not clear. What are the bands to compare? It seems that 10 μ M of WT generated less bands at 76 and 102 kDa. Could it be that the labeling is swapped?

- This assay was to measure the selective competition for USP18 by the Agb probe. When lysates are preincubated with Ac-mISG15^{CTD}[WT]-PA, the active site of ISG15-reactive DUBs is occupied by this covalent probe and no longer binds to Cy5-mISG15^{CTD}[WT]-PA thus decrease in fluorescent signals is observed after labeling with the Cy5 WT probe for all ISG15-reactive proteins. When lysates are preincubated with Ac-mISG15^{CTD}[R153Agb]-PA, however, the active site of USP18 is more selectively occupied by the probe over other DUBs and other ISG15-reactive DUBs are still available to bind to Cy5-mISG15^{CTD}[WT]-PA, resulting in the selective decrease in fluorescent signal from USP18 is observed.

32. Sup Fig 6C suggest that the mutant ISG15-Agb binds less to USP18 compared to WT USP18 (top right panel UB: USP18)? However, this data clearly demonstrates that no more USP14 binds to ISG15-Agb. This is a strong compelling case for the specificity of ISG15-Agb to USP18. However what about USP5 (tested in sup Fig 6A-B or the others DUBs?). Although this question is addressed in Sup Fig 7B. it would be nice to confirm this by Western blot. Additionally, if the authors test USP14 by western-blot and USP5 by in vitro assay in the sup fig 1, they should be consistent and always test SUP5 and USP14 across the whole manuscript.

- The figure now includes USP5 western blot data.

33. Fig 3D and sup Fig 3B are strong evidence for the enhanced specificity of ISG15-Agb. This is very compelling. However, as the authors mentioned, the expression level of the DUBs plays a role in the binding to the probe. Did the authors consider analyzing the proteome of HEK293 cells +/- USP18 overexpression to check for the expression level of the DUBs and USP18? Alternatively, the authors could perform a competition assay in vitro to check whether at the same concentration, ISG15-Agb binds more to USP18. They could mix all the DUBs: USP5, USP14, USP16 and USP18 at equimolar concentration and check which DUBs bind most to ISG15 WT versus Agb.

- Comparative analysis of expression levels of different proteins is not quite accurate due to limitations of the detection method. We acknowledge that the ability and sensitivity of these probes to detect USP18 or other DUBs may depend on the relative expression and activity of these proteins in the conditions used. However, as we have shown that R153Agb mutant probes, with enhanced specificity for USP18 compared to WT ISG15-based probes, can be used to profile the activity of USP18 in physiologic conditions where these DUBs are differentially expressed, especially when used in combination with pan-DUB Ub-based probes, we do not think it is necessary to assess which DUBs bind most to these probes when they are equally present.

34. "As expected, USP18 was the protein most highly enriched by the R153Agb probe followed by USP16 (Fig. 3D)" while USP5 and USP14 do not bind to ISG15Agb, USP16 seems to retain binding Fig 3D. Is this an artefact due to high expression of USP16 in HEK293 cells? May be this is less of an issue in other cell lines. The authors should address the remaining reactivity of ISG15-Agb to USP16.

- This has been addressed. Please see the response above.

35. "Overall, the ABPP data recapitulate the selectivity profile of our USP18 ABP and establish its utility in quantitative chemoproteomics as a chemical tool to selectively report USP18 activity in various cell lines." The authors demonstrated that in cell lysate overexpressing USP18 the ISG15-Agb is more specific for USP18. However, this is dependent on the expression level of USP18 and the other DUBs. The authors should repeat this at a physiologic level of USP18 such as in A549 cells or other cell lines. It would be nice to check the expression level of USP18 along with USP2, USP5, USP14, and USP16. Even if challenging, the expression level should be analyzed by MS but also by Western blot (although comparing the expression level by Western blot is far less accurate since the detection relies on different primary antibodies).

- This has been addressed. Please see the response above.

36. Figure 3D, other proteins are also enriched as strong as overexpressed USP18. The authors should comment on those. What are they? Also DUBs or other proteins?

- Other proteins enriched by the R153Agb probe compared to DMSO (\log_2 ratio > 3) were CPVL, DCAF8, NBAS, and TUBB8. However, their MS intensity was significantly lower than that of USP18 (Supplementary Table 2).

37. Sup Fig 8 shows that PR-619 also inhibits USP18 activity similar to WP1066, why not studying this inhibitor as well?

- PR-619 is a broad-spectrum DUB inhibitor that has been widely employed as a tool to investigate the role of ubiquitination in various cellular processes.

38. Fig 4D, nice data showing that ISG15-Agb is more specific for USP18. And, that WP1130 inhibits USP18 only but not the other DUBs (as can be seen with WT ISG15). However, the blot showing USP18 expression (loading/expression control) is lacking. As mentioned above, it would also be nice to show that the bands disappearing are actually USP5, USP14 or USP16. This could be readily performed with a western blot analysis.

- The figure now includes USP18 western blot data.

39. Fig 4E, the data nicely shows the dose dependence effect (difference between 100 vs 50) but it also shows that WP1130 does not inhibit ubiquitin-specific DUBs as most bands remain with the same intensity. This is also shown with the blot against USP14 showing that WP1130 does not inhibit USP14 binding to ubiquitin. However, the authors mentioned above that WP1130 was found to inhibit USP14 activity towards ubiquitin: "However, we found that several reported DUB inhibitors, including WP1130 (a known inhibitor of USP5, USP9X, USP14, USP24, and UCH37, Fig. 4B)" However, the data in Fig 4E shows that the reverse where actually USP14 is not inhibited by WP1130. The authors should comment on this.

- The following sentence has been added. "The reason we were unable to detect other DUBs that WP1130 has been known to target in this experiment might be due to the use of USP18 overexpressing lysates, as we observed WP1130 competitively blocks labeling of USP9X and USP24 in HEK293T cell lysates (Supplementary Fig. 10C and Supplementary Table 3), and the reversible nature of WP1130 binding to targets wherein the outcome of competition assay largely depends on the experimental conditions used."

40. Fig 4F-G, this is a very elegant approach to investigate cross-reactive inhibitors. Additionally, the authors could show all the identified DUBs as a heatmap, this will help to appreciate that USP18 is the most inhibited while few other DUBs are also inhibited by WP1130.

- As mentioned, WP1130 showed a quite narrow reactivity scope with significant binding to 4 DUBs including USP18. The heatmap analysis of selected DUBs was included in Supplementary Fig. 10B and a list of DUBs enriched by probes is found in Supplementary Table 2.

41. Fig 4F-G, dots are colored according to the DUB family. This should be mentioned in the legend or on the graph. The dots corresponding to the DUBs could be bigger to help distinguish the colors.

- The legend has been modified.

42. Sup Fig 9B the scale is indicated but it is not clear what are the values about. Is this fold change, intensity or z-score?

- It is a \log_2 fold change of MS intensity.

43. "Although not specific, we thus report WP1130 as a promising, first example of small-molecule USP18 inhibitors" is that true? No this is not true. The first small molecule inhibiting USP18 protease activity over ISG15 was reported already [PMID: 25639862]. The authors should mention the paper. The authors could claim that WP1130 is the first strong inhibitor of USP18 but this should be supported with data showing the IC50.

- The following sentence has been added to clarify. "Of note, a previously reported USP18 inhibitor contains α,β -unsaturated dienone with two sterically accessible electrophilic β -carbons,⁵⁰ similar to b-AP15⁵¹ and RA190^{52,53}, acting as a Michael acceptor to target nucleophiles including catalytic cysteine of several isopeptidases with potential chemical liability and promiscuity problem. Thus, we chose WP1130 (a known inhibitor of USP5, USP9X, USP14, USP24, and UCH37, Fig. 4B),⁵⁴⁻⁵⁹ which showed concentration-dependent inhibition of USP18 with an IC50 value of 25 μ M (Supplementary Fig. 8C), to further characterize in chemoproteomic assays."

44. "Nonetheless, our study demonstrates that utilizing this USP18 ABP in conjunction with a Ub-based DUB probe in a chemoproteomic screening platform holds promise as a tractable workflow for global, competitive profiling of DUB inhibitors to perform simultaneous hit identification and selectivity assessment" I fully agree with this statement. Additionally, this can be done in a cell lysate compared to other state-of-the-art approach. One perspective would be to perform this screen in cellulo. However, crossing the cell membrane might be challenging although several methods have been envisioned and tested.

- We agree and our ongoing and future studies include further optimization of the probes to enable intracellular profiling of USP18 activity.

45. "Although we used the USP18 overexpressing system as a model, this screening approach can be applied to native or disease models of interest." Indeed, it is necessary to use a more relevant model without overexpression. Alternatively, the expression level of USP18 in HEK293T cells overexpressing could be lowered by transfecting less plasmids. It would be worthwhile to show whether HEK293T cells overexpressing USP18 express USP18 close to the physiologic level.

- This has been addressed. Please see the response above.

47. "A549, H358, H2030, and H1650 lung cancer lines were selected and first evaluated for their USP18 expression" abbreviation of "interferon" to "IFN" is missing. The authors could mention the concentration and time of interferon treatment this is relevant to evaluate whether the interferon treatment was performed with experimental conditions similar to physiologic level. Ideally this is mentioned in U/mL.

- It has been mentioned in the figure legend.

48. Fig 5B, the WB against USP18 is essential to evaluate the effect of ISG15-Ag. Again, here the authors should show quantification of the western blot. The quantification of the Cy5 is shown on the right side but it would be better and more specific to show the same quantification using the western-blot data against USP18. Additionally, USP5, USP14, and USP16 could also be tested by western blot. When relevant the experiments should be performed with replicates to evaluate the distribution of the data.

- We used a more robust MS-based ABPP approach to identify and quantify reactive DUBs and the new data are now included in Fig. 5C and Supplementary Figs. 13-14 and Supplementary Table 4)

49. "The fluorescent intensity of the USP18sf (truncated isoform)-probe conjugate band measured by densitometry indicated a slightly higher activity of USP18 in H2030 cells" the authors should indicate "USP18sf" in the figure.

- This has been corrected.

50. What is the difference between Fig 5 B and sup fig 11? Is this an independent experiment repeat?
- Fig. 5B is the data with the R153Agb probe and Supplementary Fig. 11 (now 12) is the data with the WT probe.

51. "...in which USP18 is highly activated and provide insights into posttranslational regulation of USP18" the authors should explain and may be showcase how information about USP18 PTM can be retrieved with their approach. They should show more data.

- The sentence has been modified to “We reason this approach of measuring the differential activity of USP18 combined with expression analysis can help investigate cancer types and states in which USP18 is highly activated and provide insights into the cellular context in which USP18 inhibitors can have translational potential.”

52. Figure 5 could also show whether the probe ISG15-Agb reacts with USP5, USP14 and USP16 (+/-IFN). This should be tested with a “high exposure” western blot against USP18 like in Fig 5A middle panel. It would also be nice to test the WP1130 in those cell lines as well.

- The chemoproteomics analysis in A549 cell lysates prepared after IFN stimulation is now included in Fig. 5C-D, Supplementary Figs. 13-15 and Supplementary Tables 4-5. Please see the response above.

Response to reviewers' comments

We appreciate the reviewers' comments. It is gratifying that reviewers find our revised manuscript improved.

Enclosed is a summary of the key changes, a point-by-point response to Reviewer #2's comments, and the revised manuscript and supporting information. Overall, we believe these changes address the reviewer's concerns and have strengthened the manuscript.

• Determining the reactivity of the mISG15_{CTD}[R153Agb] probe with USP16:

1) The authors demonstrated that the ISG15 R153Agb mutant binds less to USP16 compared to WT ISG15, but they failed to demonstrate that USP18 binds more to the ISG15 R153Agb mutant compared to WT ISG15. Data showing the contrary would be crucial to support the main claim of the paper. The authors could determine the affinity of USP18 and USP16 for WT ISG15 and ISG15 R153Agb mutant. Alternatively, the authors could design a chemoproteomics screen to resolve the issue of this dual specificity toward USP18 and USP16. One could propose to deplete the cell lysate of USP16 by immunoprecipitation or genetic engineering (KO cells?). Using such an approach, the ISG15 R153Agb mutant probe would be fully specific to USP18.

>> We do not claim that USP18 binds more to the ISG15 R153Agb mutant compared to WT. Our data rather support that labeling of other ISG15 cross-reactive DUBs by ISG15 R153Agb mutant is decreased compared to WT while labeling of USP18 by WT and Agb mutant is comparable. Following the reviewer's suggestion, we performed an additional experiment to provide an alternative chemoproteomics strategy for selective detection of USP18. But, instead of immunoprecipitation or genetic engineering, we designed a competitive chemoproteomics method where cell lysates are preincubated with a Ub-based covalent inhibitor prior to labeling with ISG15-based probes to block DUBs that react with both Ub and ISG15 such as USP16, USP5, and USP14. New data are now included in Supplementary Fig. 15 and the results are described as follows:

"To absolutely avoid the potential complications of cross-reactivity of the probes with USP16, one can design an alternative chemoproteomics strategy where all the Ub-reactive DUBs are blocked with a Ub-based covalent inhibitor prior to labeling with ISG15-based probes. To test this, we performed a labeling assay with fluorescent probes in A549 cell lysates. When we incubated lysates of IFN-stimulated A549 cells with Cy5-mISG15_{CTD}[R153Agb]-PA, we could again detect USP18 with decreased labeling of USP14, USP5, or USP16 (estimated based on the molecular weight) compared to WT probe as expected (Supplementary Fig. 15). The labeling of these DUBs that react with both Ub and ISG15 was further diminished when the lysates were preincubated with Ub-PA (1 μ M, 1 h) while the labeling of USP18 was maintained. Similar competitive approaches can be applied to the MS-based detection of USP18 using biotin probes."

2) I thank the authors for the new data provided to investigate the cross-reactivity of the ISG15 R153Agb mutant toward USP16. I believe this is of clear added value and demonstrates that the ISG15 R153Agb mutant has reduced binding to USP16 compared to WT ISG15. However, this data (Fig 7) fails to demonstrate the superior specificity of the ISG15 R153Agb mutant to USP18 compared to USP16. Instead, the data in Figure 5C suggests that the ISG15 R153Agb mutant has dual specificity for USP18 and USP16, likely to the same extent, and contradicts the data in Sup Fig 7. Can the authors comment on this discrepancy? How do the authors explain that USP16 binds to ISG15 R153Agb mutant and WT ISG15 to the same extent? One could suppose that the results can be explained by the higher expression level of USP16 in A549 cells compared to HEK293T. Consequently, the pulldown of ISG15 R153Agb mutant enriched the same amount of USP16 as the pulldown of WT ISG15 despite USP16 having higher affinity for WT ISG15 and less affinity for ISG15 R153Agb mutant (Fig 7). This is a well-known principle where the enrichment of prey protein (here USP16 or USP18) by pulldown is driven by affinity to the bait (here ISG15 R153Agb) but also by the concentration of the prey. It would be worthwhile to determine the abundance of

USP16 and USP18 by shotgun proteomics and IBAQ intensity to explain the different results in Fig 5C and sup Fig 7.

>> The difference between the data presented in Fig. 5C and Supplementary Fig. 7 is that we used 1 μ M of probe in HEK297T cell lysates (Supplementary Fig. 7) whereas 5 μ M of probe was used in IFN-stimulated A549 cell lysates (Fig. 5C). The reason for using a higher concentration was to ensure sufficient enrichment of USP18 when the level of protein expression is relatively low. As we increased the probe concentration, we detected more of USP16 along with USP18 indicative of concentration dependency. The optimal concentration of the probe for selective detection of USP18 can vary depending on the cell types and states. As pointed out by the reviewer, relative expression levels of each active protein can determine the extent of labeling and enrichment. Unfortunately, when we conducted shotgun proteomics using iBAQ intensity to determine protein levels of USP18 and USP16 in A549 cells, we were unable to detect USP16 for a reason that is not clear to us although we could detect it by western blotting and LC-MS after enrichment (see the data below, now included in Supplementary Fig. 11B).

Instead, to assess the relative reactivity of ISG15-based probes between USP18 and USP16, we incubated an equimolar mixture of recombinant human USP18 and USP16 with AC-mISG15_{CTD}-PA probes and analyzed the extent of labeling of each protein. Both WT and R153Agb mutant probes reacted with USP18 and USP16 to a similar extent (Supplementary Fig. 20). The extent of labeling was comparatively, slightly less with the R153Agb probes at 5 μ M. As we DO NOT intend to claim that our Agb mutant probes react with USP18 with superior specificity compared to USP16, we have added sentences as follows:

“Together, these results suggest that the R153Agb mutant of mISG15_{CTD} binds and interacts with USP16 but at a comparatively lower level than the WT depending on the concentration used. The ability and sensitivity of these probes to detect USP18 or USP16 will depend on the relative abundance and activity of these two proteins in the conditions used. To avoid the potential issues of this cross-reactivity, careful considerations need to be taken in designing experiments and interpreting the results.”

“As our R153Agb mutant probes bind and interact with USP16 to some extent, our ongoing investigation also includes understanding the molecular interactions between ISG15 and USP16 that can help design more selective probes for USP18 with minimal or no cross-reactivity with USP16.”

- Investigating the functional effects of USP18 inhibition:

The inhibition of USP18 by WP1130 is clearly not supported by the data. The increase in ISGylated proteins in the insoluble fraction is more likely due to cellular stress or toxicity (or another biological phenomenon: autophagy induction through p62 or HDAC6?), rather than USP18 inhibition. In this experiment (Fig 6), the authors chose to perform the drug treatment with WP1130 on living cells, while in the previous experiments (Fig 4 and 5), they performed the treatment in a cleared cell lysate. To demonstrate USP18 inhibition by WP1130, the authors could also perform the drug treatment in cell lysate and monitor delISGylation activity

by western blot to detect ISGylated proteins. This has been done elsewhere using catalytically inactive USP18 versus wild-type (PMID: 28165509, Figure 2C). Showing this data will demonstrate functional effects at the molecular level but not at the cellular level indeed. Functional effects at the cellular level could be addressed in future work.

>> We thank the reviewer for pointing this out. We performed an additional experiment where lysates of A549 cells stimulated with IFN were incubated with WP1130 in the absence or presence of recombinant human USP18. The new data are added in Fig. 6 and Supplementary Fig. 17 and the results are described as follows:

“To analyze the effect of WP1130 on protein ISGylation, lysates of A549 cells stimulated with IFN- β were incubated with WP1130 for 2 h. No significant change in ISGylated proteins was observed (Fig. 6A). When the cell lysates were incubated with recombinant human USP18 (5 μ M), a decreased level of ISGylated proteins was detected indicative of deISGylating activity of USP18, which was partially blocked in the presence of WP1130 suggesting that WP1130 inhibits the protein deISGylation. On the other hand, its effect on protein ubiquitylation was not clear (Supplementary Fig. 17A). In A549 cells, WP1130 displayed potent cytotoxic (Supplementary Fig. 18A) and antiproliferative properties (Supplementary Fig. 18B). Next, A549 cells were stimulated with IFN- β for 48 h to induce USP18 expression and cellular ISGylation and treated with increasing concentrations of WP1130 for 2 h. Cell lysates were then analyzed by immunoblotting. We decided to separate detergent-soluble and insoluble fractions since previous studies have shown that WP1130 treatment leads to rapid accumulation of ubiquitylated proteins in the detergent-insoluble fraction indicative of aggresome formation through DUB inhibition.^{57,65} In agreement with that, we observed the accumulation of ubiquitylated, insoluble aggregates of protein in WP1130-treated cells (Supplementary Fig. 17B). Similarly, ISGylated proteins were significantly accumulated in the insoluble fraction upon WP1130 treatment in a dose-dependent manner (Fig. 6B). Interestingly, WP1130 treatment increased USP18 levels detected in the insoluble fraction while a mild decrease was noted in the soluble fraction. It has been reported that ISG15 conjugation marks target substrates for interaction with histone deacetylase 6 (HDAC6) and SQSTM1/p62 toward autophagic clearance.⁶⁶ Our study suggests that WP1130 treatment enhances cellular levels of ISGylated proteins that can trigger aggregate formation (aggresomes). Understanding whether these cellular effects of WP1130 observed are direct consequences of USP18 inhibition and if USP18 itself is directly associated with these aggregates warrants additional investigation.”

Minor comments:

- For the activity-based probe profiling (ABPP) in lysate, the reaction volume should be mentioned. The method section only contains the protein amount and probe concentration. This information is important if one wishes to repeat the assay.

>> The reaction volume has been provided in the Methods section.

- Regarding the use of USP18 ABP under physiological conditions: “One of the advantages of using the probe more specific for USP18 in the ABPP approach when combined with Ub-based ABP is that we avoid significant competition between probes for DUB profiling,” said by the authors, but this strategy is unclear. One alternative approach would be to deplete the lysate of “ubiquitin-binding DUBs” with the ubiquitin probe as a first step. Then, in a second step, this “ubiquitin-cleared” lysate could be used for the mISG15ctd[R153Agb] probe. The authors should comment on this strategy or mention this alternative in the text.

>> The sentence was deleted in the revised manuscript. We now comment on the alternative chemoproteomics strategy as follows:

“To absolutely avoid the potential complications of cross-reactivity of the probes with USP16, one can design an alternative chemoproteomics strategy where all the Ub-reactive DUBs are blocked with a Ub-based covalent inhibitor prior to labeling with ISG15-based probes. To test this, we performed a labeling assay with fluorescent probes in A549 cell lysates. When we incubated lysates of IFN-stimulated A549 cells with Cy5-mISG15_{CTD}[R153Agb]-PA, we could again detect USP18 with decreased labeling of USP14, USP5, or USP16 (estimated based on the molecular weight) compared to WT probe as expected (Supplementary Fig. 15). The labeling of these DUBs that react with both Ub and ISG15 was further diminished when the lysates were preincubated with Ub-PA (1 μ M, 1 h) while the labeling of USP18 was maintained. Similar competitive approaches can be applied to the MS-based detection of USP18 using biotin probes.”

- The authors said, “Although R153Agb mutant probes are more specific for USP18 over other ISG15-crossreactive DUBs compared to existing WT ISG15-based probes.” What are the data supporting this claim? To support this statement, the authors should report binding affinity (K_d, K_{on}, or K_{off}). Alternatively, the authors can rephrase this statement. While the authors show that the R153Agb mutant probe is less reactive to USP5 and other DUBs compared to WT ISG15, they did not show that the R153Agb mutant is more specific for USP18 compared to other DUBs. There is a subtle difference in the phrasing. This was already mentioned above as a major comment but please re-phrase the sentence.

>> The sentence has been rephrased to “Although R153Agb mutant probes are less reactive to USP5 and USP14 compared to WT ISG15-based probes,..”

- The authors said, “Our chemoproteomics data indicated that they react with USP16 to some extent, especially when USP18 is expressed at a relatively low level.” This statement should be supported with some data. Is this referring to the expression level of USP18 in A549 cells (Fig 5C)? The expression level of USP18 seems unknown from the manuscript. Could the authors show the expression level of the DUBs in the lysate of A549 +/- interferon? (and HEK293T cells?) The authors could perform shotgun proteomics on those samples and show protein rank as before. Expression level can be evaluated using iBAQ intensity values. Although not perfect, iBAQ intensity allows an estimation of protein expression and comparison between proteins. Alternatively, the authors could make use of proteomics or transcriptomics data already published elsewhere.

>> The analysis of the relative abundance of USP18 determined by shotgun proteomics using iBAQ intensity is now included in Supplementary Fig. 11B. The abundance of USP18 in IFN-stimulated A549 cells was found to be lower than other ISG15 cross-reactive DUBs such as USP14 and USP5.

- Figures 7B and 7C should be quantified and represented with a bar plot.

>> A bar plot has been added to Figures 7B and 7C.

- The sentence here is unclear please rephrase: "Comparing other proteins detected by R153Agb relative to the WT, USP16 was found to be ranked high while enrichment of USP5, USP14, USP30, and USP35 was significantly diminished, indicating the R153Agb probe reacts with USP16 when its target protein USP18 is expressed at a relatively low level".

>> The sentence has been rephrased to “The enrichment of USP5 and USP14 by R153Agb probe was reduced relative to the WT probe. However, USP16 was pulled down by both probes at a similar extent, indicating the R153Agb probe reacts with USP16 when its target protein USP18 is expressed at a relatively low level.”

- In Fig 5D, the legend should be updated to clarify that the heatmap was made based on results from mixed probes (ISG15 probe and Ub probe). This is clearly stated in the text, but not in the legend.

>> The legend has been modified to state that the heatmap analysis was done on the results from competitive profiling using mixed probes.

Response to reviewers' comments

I thank the authors for performing additional experiments. Almost all my previous comments have been addressed. I have one final minor comment (see below). Upon completion, I recommend publication of this manuscript.

>> We appreciate the reviewer's comments. It is gratifying that the reviewer finds our revised manuscript improved.

In their answer, the authors indicate that they do not claim that the ISG15Agb has more specificity for USP18 over USP16. But, this seems to be essential for designing an ISG15 probe specific for USP18 over other DUBs (e.g. USP16) as mentioned in the title. How do the authors intend to study exclusively USP18 then? To do so, one can increase the affinity of the ISG15 probe for USP18. Alternatively, one can deplete the ISG15-cross reactive DUBs by genetic knock-out or immunoprecipitation. The authors proposed to covalently block all ubiquitin DUBs prior using the ISG15 probe (sup fig 15). This another elegant strategy to reach the same goal. This approach is convincing however the sup fig 15 is lacking the immunoblotting for the ISG15 cross-reactive DUBs, especially USP16. The indication of the molecular weight is not sufficient.

>> Thanks for pointing this out. The reason why we did not include immunoblotting results in supplementary figure 15 for the ISG15 cross-reactive DUBs is that we already have shown that our ISG15 probe does not react with USP5 and USP14 to the extent that can be detected by immunoblotting. Also, as mentioned in the manuscript, the USP16–probe conjugate was not detected by the anti-USP16 antibodies we purchased and tested. To address the comment, we have added the following in Discussion. “Since our R153Agb mutant probes bind and interact with USP16 to some extent, our ongoing investigation also includes understanding the molecular interactions between ISG15 and USP16 that can help design more selective probes with increased affinity for USP18 and minimal or no cross-reactivity with USP16. Alternatively, to avoid the potential complications of cross-reactivity, one can deplete the ISG15 cross-reactive DUBs by genetic knockout or immunoprecipitation or design a chemoproteomics method where all the Ub-reactive DUBs are blocked with a Ub-based covalent inhibitor prior to labeling with ISG15-based probes. As we confirmed the feasibility of the latter strategy in a gel-based labeling assay, we anticipate the combination of strategies provided in this study, particularly, utilizing MS-based quantitative proteomics analysis can help investigate functions of USP18 more exclusively in relevant biological settings.”